# Towards Positive Geometries of Massive Scalar field theories

**Mrunmay Jagadale,**[a] **Alok Laddha,**[b]

[a]*California Institute of Technology, Pasadena, CA 91125, USA*
[b]*Chennai Mathematical Institute, Siruseri, Chennai, India.*

*E-mail:* mjagadal@caltech.edu, aladdha@cmi.ac.in

ABSTRACT: Building on the prior work in [1] we locate a family of positive geometries in the kinematic space which are a specific class of convex realisations of the associahedron. These realisations are obtained by scaling and translating the kinematic space associahedron discovered by by Arkani-Hamed, Bai, He and Yan (ABHY). We call the resulting polytopes, deformed realisations of the associahedron.

The deformed realisations shed new light on the CHY formula. One of the striking discoveries in [2] was the fact that the CHY scattering equations generate diffeomorphism between the (compactified) CHY moduli space $\mathcal{M}_{0,n}(\mathbb{R})$ and the ABHY associahedron. As we argue, the deformed realisation of the associahedron can also be interpreted as an diffeomorphic image of $\mathcal{M}_{0,n}(\mathbb{R})$ under a class of scattering equations that we call deformed scattering equations. The canonical form in the kinematic space is thus once again the push-forward of the Parke-Taylor form . A natural off-shoot of our analysis is the universality of the Parke-Taylor form as a CHY Integrand for a class of (tree-level and planar) multi-scalar field amplitudes.

These ideas help us in proving the existence of positive geometries for certain specific multi-scalar interactions. We prove that in a field theory with a massless and a massive $(\phi_1, \phi_2)$ bi-adjoint scalar fields which interact via, $\lambda_1\phi_1^3 + \lambda_2\phi_1^2\phi_2 + \lambda_3\phi_1\phi_2^2$ interaction, the tree-level S-matrix with massless external states expanded up to $\lambda_2^2$ is a weighted sum over Canonical forms defined by certain deformed realisations of the associahedron. Finally, we show that these ideas admit an extension to one-loop. In particular, the one loop S-matrix integrand upto $O(\lambda_2^2, \lambda_3)$ is a weighted sum over canonical forms of a family of deformed realisations of the type-D cluster polytope, discovered in [3, 4].

## 1   Introduction

The Amplituhedron program ( [5], [6], [7] and references therein) is an attempt to classify S-matrix in terms of differential forms associated to a specific class of polytopes such that the irreducible postulates of the old S matrix program, such as unitarity and locality are derived from certain deeper principles which are intrinsically tied to the geometry and combinatorics of these polytopes. In the context of planar Super-Yang-Mills and massless bi-adjoint scalar field theories, there is a strikingly coherent picture that continues to emerge.

From a minimal structure of kinematic space (constrained only by Poincare invariance of flat space), and under the assumption of color-ordering, one discovers a class of polytopes located inside the kinematic space. All of these polytopes belong to a class known as positive geometries and each positive geometry induces a unique differential form on the (projectivized) kinematic space. In the seminal paper [2], Arkani-Hamed, Bai, He and Yan (ABHY) discovered specific kinematic space realisation of a combinatorial polytope known as associahedron such that when the canonical form defined by the associahedron is restricted onto the ABHY realisation, one obtains scattering amplitudes for a bi-adjoint scalar theory with cubic interaction.

Even at this relatively early stage of the program, following questions naturally emerge.

- What are the class of positive geometries whose associated forms are the S-matrix of some theory ?

- Given a combinatorial polytope, why does a very specific convex realisation of the polytope generates scattering amplitude. A combinatorial polytope such as associahedron has an infinitude of convex realisations, and how many of them are associated to QFT S-matrix?

These questions are tied to a remarkable result proven in [8, 9] which can be summarised as follows. Out of an infinity of convex realisations of an associahedron (and in general any accordiohedron) in $\mathbb{R}^{\frac{n(n-3)}{2}}$, there is a family of convex realisations which are polytopal realisations of quiver of type $A_n$ (or in general a dissection quiver). And it is precisely these realisations that generate S-matrix of massless scalar field theories. It is this striking connection between algebraic combinatorics (and more specifically the theory of gentle algebras), associahedron (Accordiohedron) polytope and S-matrix of a local QFT which forms the basic edifice of the S-matrix program .

The robustness of this edifice makes us wonder if there is any "space" to obtain a S-matrix of massless scalar particles with generic scalar interactions involving a spectra of scalar fields with different masses. However one juncture at which an "additional" input has been put in these constructions is the identification of $\mathbb{R}^{\frac{n(n-3)}{2}}$, the embedding space in which associahedron and accordiohedron is realised from the polytopal fan construction [2, 8, 9], with the kinematic space $\mathcal{K}_n$. In this paper, we explore some of the simplest consequences of scrutinizing and relaxing this input. That is, we consider a class of linear maps from the embedding space in which an associahedron (or more generally an accordiohedron is realised) to the kinematic space of scalar particles. As we show, the result is a class of positive geometries in $\mathcal{K}_n$ which are deformed realisations of the associahedron (accordiohedron) and for a class of deformations these are positive geometries for S-matrix of multi-scalar field theories in which the scalars have unequal masses.

More in detail, if the map between the embedding space and kinematic space is not identity, then the ABHY realisation of an associahedron in the embedding space appears deformed in the kinematic space. We will refer to all such realisations as deformed associahedron. Each

deformed realisation is a simple polytope and defines a canonical form in the (projectivized) Kinematic space. In this paper, we initiate an investigation into the conditions under which the canonical form defined by the deformed associahedron is the (color-ordered) tree-level amplitude of a local quantum field theory. Although a complete analysis of the relationship between deformations and QFT amplitudes is beyond the scope of this work, for several classes of deformations, we write down the interactions whose scattering amplitudes are the canonical forms. In a nut-shell the picture that emerges can be summarised as follows.

| Massive poles | $\longrightarrow$ | Shift the positive wedge |
|---|---|---|
| Multiple fields | $\longrightarrow$ | Rotate the hyper-planes |
| Higher-point interactions | $\longrightarrow$ | Project the polytope |

Next, we ask the "reverse" question. Given a Lagrangian consisting of several bi-adjoint scalars with distinct masses and cubic interactions between different fields, what are the deformations of ABHY associahedron which generate the corresponding S-matrix. Once again, we analyze this question in the simplest possible example.

Following our earlier work [1], we consider a Lagrangian consisting of two scalars $\phi_1$, $\phi_2$ with unequal masses and a cubic interaction $\lambda_1 \phi_1^3 + \lambda_2 \phi_1^2 \phi_2$. We prove that a specific deformation of the ABHY associahedron is the positive geometry of this perturbative S-matrix. This example is motivated by our previous work [1] in which we had argued that a family of associahedra denoted as $A_{n-3}^{(i,j)}$ whose co-dimension one facets are located at

$$X_{kl} = m^2 \text{ if } (k,l) \in \{ (i,j), \ldots, (i+|j-i|-1, j+|j-i|-1) \} \tag{1.1}$$

generate S-matrix of the two-scalar field theory to $O(\lambda_2^2)$. More in detail, it was proved that a certain weighted sum over $A_{n-3}^{(i,j)} \forall (i,j)$ along with. $A_{n-3}$ (the ABHY associahedron where all facets correspond to massless poles) produces the desired S-matrix. In [1], such translated associahedron was referred to as the colorful associahedron.

Colorful associahedron has two kinds of vertices. One class of vertex is adjacent only to $X_{ij} = 0$ facets and another class is adjacent to $n-3$ facets out of which precisely one facet is at $X_{kl} = m^2$. One drawback of our previous analysis was that residue of the canonical form defined by $A_{n-3}^{(i,j)}$ on both type of vertices was one. This is rather unnatural as the vertex of the first type is associated to a channel with purely massless poles and contributes to the S-matrix at order $\lambda_1^{n-2}$ and the second type of vertex which corresponds to a channel in which precisely one pole is massive should contribute at $\lambda_1^{n-4} \lambda_2^2$.

This drawback is naturally resolved as our deformations of ABHY realisation include a scaling in addition to the translations parametrized in eqn.(1.1). We show that given any $\frac{\lambda_1}{\lambda_2}$, there is a unique choice of $\frac{\alpha_1}{\alpha_2}$ which generates the perturbative S matrix.

The simple observation of exploring space of isomorphisms between embedding space for the ABHY associahedron and the kinematic space has interesting ramifications for the CHY formula. The non-trivial identification between kinematic space and the embedding space for polytopes can also be interpreted as deforming the CHY scattering equations so that

the "deformed" diffeomorphism between worldsheet associahedron and the kinematic space associahedron can be used to push forward the *park-taylor* form to the kinematic space. Hence the Parke-Taylor form on worldsheet can generate a wide class of scalar field amplitudes with cubic interactions where information about the internal as well as the external masses and couplings is inside the scattering equations. The CHY formula for a massive $\phi^3$ amplitude that was obtained by Dolan and Goddard is the simplest such example of deformed scattering equations.

This paper is organised as follows.

In section 2 , we review the positive geometry of bi-adjoint scalar field theories with mass $m$ and monomial interactions $\phi^p$.

In section 3 we parametrize the embedding space $\mathbb{R}^{\frac{n(n-3)}{2}}$ which admit the type cone realisations of associahedron and in general accordiohedron in terms of the action of $\mathcal{G}$ on the kinematic space $\mathcal{K}_n$ of massless particles.[1] For a generic action of $\mathcal{G}$ we obtain a deformed realisation of the associahedron in the kinematic space.

In section 4 we show how the canonical form defined by such a deformed realisation is the push-forward of Parke-Taylor form by scattering equations which generate diffeomorphism between worlsheet and the embedding space. In section 4.3, we illustrate how deformations can lead to S-matrix of a local QFT. In particular, we analyse a specific class of deformed realisation $A_{n-3}^{\{\alpha\}}$ where the deformation parameters are not arbitrary but are labelled by the "shortest distance" between two vertices of the polygon. As we show $A_{n-3}^{\{\alpha\}}$ is a positive geometry for an scattering amplitude involving $n$ massless states and an interaction parametrized by the deformation parameters $\{\alpha\}$.

In the following section, 5, we start with a specific interaction and prove that there exists a family of deformed realisations such that a weighted sum over the corresponding canonical forms is the S-matrix upto a given order in perturbation theory.

Finally in sections 6-6.2, we show how these results naturally extend to one-loop. In particular after reviewing the construction of the so-called $\hat{D}_n$ cluster polytope in section 6, which is a positive geometry for 1-loop bi-adjoint scalar $\phi^3$ planar integrand, in the subsequent sections we extend the results of 5 to one loop case. We finally end with a discussion of the results in section 7.

## 2   Review of the ABHY associahedron, massless S-matrix and accordiohedron projections

In this section, we give a brief overview of the amplituhedron picture for planar scalar field theories at tree level. For more details and gerenral overview of the subject we refer to [1, 2, 10–15]. The poles of tree level scattering amplitude in a planar theory are of the form $\frac{1}{X_{ij}-m_{ij}}$, where $X_{ij} = (p_i + p_{i+1} + \cdots + p_{j-1})^2$, with $p_i$ being the momenta of the $i$th external

---

[1]Throughout this paper, $\mathcal{K}_n$ will be the "physical" kinematic space as we will analyse S-matrix of massless external particles with different interactions

particle. Not all poles can occur in a single term of the amplitude. For example, at 5-point, the poles $\frac{1}{X_{13}-m_{13}}$ and $\frac{1}{X_{24}-m_{24}}$ can not occur together in the same term. This gives the set of all poles possible in a tree level $n$-point scattering amplitude of a planar theory with one scalar field, a structure of a combinatorial polytope. This combinatorial polytope is called associahedron.

Not all poles that are possible in a tree level $n$-point scattering amplitude of a planar theory can occur in all theories. For example, the pole $\frac{1}{X_{13}-m_{13}}$ would never occur in a theory with only quartic interaction. On the other hand all poles that are possible in a tree level $n$-point scattering amplitude of a planar theory do occur in the theories with a cubic interaction. Thus positive geometry of a theory with a cubic interaction play a central role in the world of positive geometries of scalar theories.

We will now focus on planar massless $\phi^3$ theory. To get the scattering amplitude of this theory we have to realise this combinatorial polytope in the kinematic space. We represent the kinematic space of scattering momenta for $n$-particle scattering amplitudes in which all the external states are scalars by $\mathcal{K}_n$ [2]. In $D > n$ dimensions $\mathcal{K}_n$ is co-ordinatized by linearly independent planar kinematic variables $X_{ij}$ with $|i-j| \geq 2$. That is, the set $\{\, X_{ij}\,|\,|i-j| \geq 2\,\}$ spans $\mathcal{K}_n$.

There is a one-to-one correspondence between the planar variables $X_{ij}$ and the diagonals of an $n$-gon. Under this correspondence each triangulation corresponds to a term in the tree-level amplitude of planar $\phi^3$ theory. This correspondence provides us a convenient way to talk about combinatorics of poles of scattering amplitudes.

To get the geometric realization we have to first consider a positive wegde whose boundaries correspond to the poles of scattinging amplitudes. This wedge is given by

$$X_{ij} \geq 0 \qquad \forall (i,j) \text{ such that } |i-j| > 1. \tag{2.1}$$

Then, we consider a set of hyper-planes that capture the combinatorics of the poles. For every triangulation of $n$-gon we have a set of hyper-planes. Given any triangulation $T$ consider the intersection of set of hyper-planes given by,

$$X_{ij} + X_{i+1\,j+1} - X_{i\,j+1} - X_{i+1\,j} = c_{ij} \qquad \text{for all } (i,j) \notin T^c, \tag{2.2}$$

where $T^c$ is the triangulation that is obtained by rotating $T$ by $\frac{2\pi}{n}$ in counter-clockwise direction. The intersection of the positive wedge and these hyper-planes gives us a ABHY realisation of associahedron.

There is a projective form associated with associahedron. This form is given by

$$\Omega\,[\mathcal{A}] = \sum_{v \in \mathcal{A}} \text{sgn}(v) \bigwedge_{(i,j) \in v} \mathrm{d}\log(X_{ij}) \tag{2.3}$$

where the sum is over vertices of the associahedron which correspond to triangulations of $n$-gon, and the wedge product is over diagonals of the triangulations. This form when restricted to the ABHY realisation of associahedron in the kinematic space gives us the scattering amplitude of planar $\phi^3$ theory.

Let's move on to tree-level scattering amplitudes of planar theories with single scalar field but now with higher-point interactions. They are given by polytopes called accordiohedra. Unlike in $\phi^3$ theory we need more than one polytope, we need a family of polytopes to capture the scattering amplitude of this theory. However they all come from the associahedron.

For each dissection $D$ of an $n$-gon there is a accordiohedron. To get the realisation of this accordiohedron we first consider a triangulation $T$ such that $D \subset T$ and consider the realisation of associahedron associated with $T$. We then project this associahedron onto the space spanned by $X_{ij}$ such that $(i,j) \in D$. This projection is a realisation of accordiohedron associated with $D$. For example, in figure 1, the green square is the realisation of accordiohedron associated with the dissection $(13, 46)$, while the red pentagon is the realisation of accordiohedron associated with the dissection $(13, 14)$.

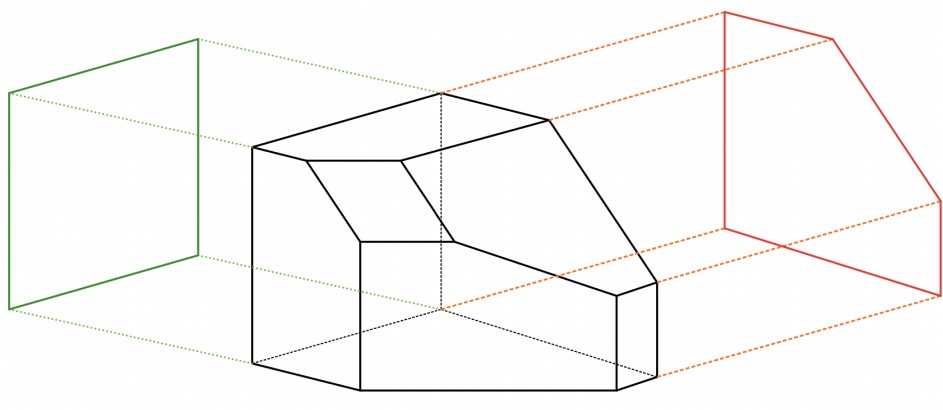

**Figure 1**: Projection of associahedron

Given the realisation of accordiohedron we consider the canonical form associated with it and then restrict it on the realisation to get contribution towards the scattering amplitude. The weighted sum of contribution from families of accordiohedra associated with dissections made of $p$-angulations with $p \in \{p_1, p_2, \ldots, p_r\}$ gives us the scattering amplitude of theory in which the interaction term has the operators $\phi^{p_1}, \phi^{p_1}, \ldots, \phi^{p_r}$.

## 3 Deforming the Planar Kinematic Variables

The kinematic space $\mathcal{K}_n$ only contains the kinematical information of the external particles. As discussed in section 2, the two main ingredients that gave us the associahedron in kinematic space (whose projective canonical form in turn gives the scattering amplitude) were the positive wedge and the hyper-planes.

| Physical input | Algebra | Geometry |
|---|---|---|
| Location of poles | $X_{ij} \geq 0$ | Positive wedge |
| Combinatorics of Scattering | $X_{ij} + X_{k\ell} - X_{i\ell} - X_{kj} = c_{ijk\ell}$ | Hyper-planes |

The equations $X_{ij} \geq 0$ tells us that in a massless scalar theory the poles of scattering amplitude are located at $X_{ij} = 0$. Therefore, this result can be trivially extended to the case when the scalar field is massive. Given a triangulation $T$, we simply consider the realisation of associahedron given by the intersection of following regions,

$$X_{ij} - m^2 \geq 0 \qquad \forall (i,j) \text{ such that } |i - j| > 1. \tag{3.1}$$

$$(X_{ij} - m^2) + (X_{i+1 j+1} - m^2) - (X_{ij+1} - m^2) - (X_{i+1 j} - m^2) = c_{ij} \qquad \text{for all } (i,j) \notin T^c. \tag{3.2}$$

This is equivalent to simply shifting $X_{ij} \longrightarrow X_{ij} - m^2$. Thus the polytope is simply the rigid translate of associahedron all of whose facets correspond to massless poles.

Given $X_{ij} \geq 0$, and $c_{ijk\ell} \geq 0$, the equations $X_{ij} + X_{k\ell} = X_{i\ell} + X_{kj} + c_{ijk\ell}$, capture the fact that, in a massless planar theory the poles $\frac{1}{X_{ij}}$ and $\frac{1}{X_{k\ell}}$ with $i < k < j < l$, can not come together in the same term of the scattering amplitude. However, the same combinatorics would be captured if we had instead consider hyper-planes given by the following equations,

$$\alpha_{ij} X_{ij} + \alpha_{k\ell} X_{k\ell} = \alpha_{i\ell} X_{i\ell} + \alpha_{kj} X_{kj} + c_{ijk\ell}. \tag{3.3}$$

With $\alpha_{ij}, \alpha_{k\ell}, \alpha_{i\ell}$ and $\alpha_{kj}$ being positive. Heuristically, one could argue that since the pole $\frac{1}{X_{ij}}$ is "no different" than $\frac{1}{X_{k\ell}}$ or any other pole, all the $\alpha$s should be same and the over all positive $\alpha$ can be absorbed into $c_{ijk\ell}$. However this begs the questions : What happens when we take the $\alpha$s to be different? We claim that taking $\alpha$s to be different gives us a realisation of associahedron that gives the scattering amplitude of theory with multiple fields.

Let us now consider the specific scenario where all the external particles have the same mass $m$.[2] We now define a new set of planar kinematic variables which are obtained from $\{X_{ij}\}$ by a linear action composed of a translation and a diagonal subgroup of $GL(\frac{n(n-3)}{2})$ with positive entries. That is consider,

$$\mathcal{G} = (\mathbb{R}_+)^{\frac{n(n-3)}{2}} \times \mathbb{R}^{\frac{n(n-3)}{2}} \tag{3.4}$$

Given a kinematic space co-ordinatized by the variables $X_{ij}$ with $X_{i,i+1} = X_{1,n} = m^2$ we define the following kinematic variables.

$$\kappa_{ij} := \alpha_{ij} \tilde{X}_{ij} = \alpha_{ij} (X_{ij} - m_{ij}^2) \tag{3.5}$$

Where $m_{i,i+1}^2 = m^2$, but for $|j - i| \geq 2$, $\alpha_{ij}, m_{ij}^2$ are arbitrary and independent positive parameters. We note that $\kappa_{i,i+1} = 0$ for massive as well as mass less external particles. We now define "deformed" Mandelstam variables $\tilde{s}_{ij}$ as,

$$\tilde{s}_{ij} = \kappa_{i,j+1} + \kappa_{i+1,j} - \kappa_{ij} - \kappa_{i+1,j+1} \tag{3.6}$$

Given any triangulation $T$, consider a set of constraints,

$$\tilde{s}_{ij} = -c_{ij} \qquad \forall (i,j) \notin T^c \tag{3.7}$$

---

[2]However we believe that ideas explored below can be generalised to S-matrix for scalar particles with arbitrary masses.

where $T^c$ is the triangulation that is obtained by rotating $T$ by $\frac{2\pi}{n}$ in counter-clockwise direction.

If we consider the ABHY realisation as a realisation of the combinatorial associahedron in an abstract embedding space $\mathbb{R}^{\frac{n(n-3)}{2}}$. Then the realisation that gives the $\phi^3$ amplitude is given by identification of this embedding space with the kinematic space such that unit vectors in $\mathbb{R}^{\frac{n(n-3)}{2}}$ get identified with $X_{ij}$. On the other hand if we identify the embedding space unit vectors with $\kappa_{ij}$ then we get the realisation above. However, now, as the embedding space is not isometric to the kinematic space (unless all the $\alpha_{ij}$ parameters are equal and non-zero), the ABHY realisation is "deformed" in $\mathcal{K}_n^+$. Here, by deformed we simply mean that the ABHY realisation is stretched or compressed in some directions. We will denote the deformed realisation as $A_{n-3}^{\{\alpha\}}$.

| Physical input | Algebra | Geometry |
|---|---|---|
| Location of poles | $\kappa_{ij} \geq 0$ | Shifted positive wedge |
| Combinatorics of Scattering | $\kappa_{ij} + \kappa_{k\ell} - \kappa_{i\ell} - \kappa_{kj} = c_{ijk\ell}$ | Rotated hyper-planes |

Let us consider an explicit example of such a deformed realisation with $A_2^{\{\alpha\}}$ with $n = 5$. If we choose the reference $T_0 = \{(13), (14)\}$ then,

$$
\begin{aligned}
\tilde{X}_{13} &\geq 0,\ \tilde{X}_{14} \geq 0 \\
\tilde{X}_{35} &= \tfrac{1}{\alpha_{35}} \left( c_{14} + c_{24} + \alpha_{14} m_{14}^2 - \alpha_{14} X_{14} \right) \geq 0 \\
\tilde{X}_{25} &= \tfrac{1}{\alpha_{25}} \left( c_{13} + c_{14} + \alpha_{13} m_{13}^2 - \alpha_{13} X_{13} \right) \geq 0 \\
\tilde{X}_{24} &= \tfrac{1}{\alpha_{24}} \left( c_{13} - \alpha_{14} m_{14}^2 + \alpha_{13} m_{13}^2 - \alpha_{13} X_{13} + \alpha_{14} X_{14} \right) \geq 0
\end{aligned}
\tag{3.8}
$$

This implies that $\tilde{X}_{13}, \tilde{X}_{14}$ are bounded by,

$$
\begin{aligned}
0 &\leq \tilde{X}_{14} \leq \tfrac{1}{\alpha_{14}} (c_{14} + c_{24}) \\
0 &\leq \tilde{X}_{13} \leq \tfrac{1}{\alpha_{13}} (c_{13} + c_{14})
\end{aligned}
\tag{3.9}
$$

Hence for any $0 < \alpha_{ij} < \infty$, one has a convex realisation of $A_2$ in $\mathcal{K}_5^+$.

The range of $\tilde{X}_{13}, \tilde{X}_{14}$ is compressed as compared to their range for ABHY realisations, where the suppression factor is $\alpha_{13}, \alpha_{14}$ respectively. As the mapping from $\mathcal{K}_n^+$ to the embedding space is simply via the positive diagonal sub-group of $GL(\frac{n(n-3)}{2})$, ABHY associahedron is mapped onto a convex realisation of associahedron for all $n$.

## 3.1   Deformed Scattering Equations and push-forward of Parke Taylor form

The CHY formula for massless bi-adjoint $\phi^3$ tree-level S matrix can be understood as the push-forward of Parke-Taylor form which is defined on the so-called world-sheet associahedron, on the ABHY realisation in $\mathcal{K}_n^+$ where the push-forward is induced by scattering equations. If the embedding space in which ABHY realisation is identified with $\mathcal{K}_n^+$ only upto $\mathcal{G}$ action, then the diffeomorphism between the worldsheet associahedron and $A_{n-3}^{\{\alpha\}}$ is via deformed scattering

equations. Which are defined as follows; Consider "scattering equations" on $\mathcal{M}_{0,n}(\mathbb{R})$

$$\sum_{j \neq i} \frac{\tilde{s}_{ij}}{\sigma_{ij}} = 0 \tag{3.10}$$

These equations are inspired by the scattering equations for mass less particles and we refer to them as deformed scattering equations. We can use these equations to generate a pushorward map from the park Taylor form on $\mathcal{M}_{0,n}(\mathbb{R})$ to $\mathcal{K}_n^+$ as,

$$\sum_{\text{solns}} \omega_n =: M_n' \bigwedge_{(ij) \in T_0} d^{n-3} \kappa_{ij} \tag{3.11}$$

From [2], $M_n'$ is given by,

$$M_n' = \sum_T \left( \frac{1}{\prod_{(kl) \in T} \alpha_{kl}} \prod_{(mn) \in T} \frac{1}{\tilde{X}_{mn}} \right) \tag{3.12}$$

We can also write the right hand side of eqn. (3.11) as,

$$\sum_{\text{solns}} \omega_n =: M_n \wedge_{(ij) \in T_0} d^{n-3} \tilde{X}_{ij} \tag{3.13}$$

where

$$M_n = \sum_T \left( \frac{\prod_{(mn) \in T_0} \alpha_{mn}}{\prod_{(kl) \in T} \alpha_{kl}} \prod_{(mn) \in T} \frac{1}{\tilde{X}_{mn}} \right) \tag{3.14}$$

In case of bi-adjoint scalar theory with mass $m$, we have

$$\tilde{s}_{ij} = 2p_i \cdot p_j + m^2 \tag{3.15}$$

The deformed scattering equations then precisely match with the scattering equations proposed by Dolan and Goddard for the massive $\phi^3$ theory.

As with the ABHY associahedron, the push-forward of the Park-Taylor form $\omega_n$ via deformed scattering equations is the canonical form on $A_{n-3}^{\{\alpha\}}$.[3] In the speical case where $\alpha_{ij} = 1 \, \forall \, (ij)$ and $m_{ij}^2 = 0, \forall \, (ij)$, we recover the canonical form on the ABHY realisation in $\mathcal{K}_n^+$.

Few natural questions arise

- Are the planar scattering forms derived in eqns.(3.11, 3.12) scattering amplitudes for a local QFT for generic choice of $\alpha_{ij}$, $m_{ij}$?

- Given a Lagrangian built out of a spectrum of (bi-adjoint) scalar fields with different masses and generic cubic interactions, is the S-matrix a (weighted sum) over positive geometries?

---

[3]The form is canonical in the sense that if we think of the kinematic space co-ordinatized by $\tilde{X}_{ij}$ as a $\frac{n(n-3)}{2}$ dimensional real Projective space, then the pushforward is the canonical form on the kinematic space with simple poles on the facets of the deformed associahedron.

We take certain preliminary steps in answering these questions in this work. More in detail,

- We classify a certain family of $m_{ij}$ and $\alpha_{ij}$ for which the deformed associahedron is an amplituhedron for *some* local quantum field theory containing a family of bi-adjoint scalars with different masses.

- As for the second question, we improve on the analysis of [1] and prove that n-point amplitude built out of cubic couplings and with one massive and $n - 4$ massless propagators is a weighted sum over canonical forms associated to certain deformed realisation of the colorful associahedra which were introduced in [1].

In the next section, we start by considering $\alpha_{ij}, m_{ij}$ configurations in increasing level of complexity. This will lead us to a family of possible deformations of ABHY associahedron such that (1) the canonical forms are pushforward of the Park-Taylor forms and (2) these forms generate a perturbative $n$-point amplitude of a local QFT.

## 4  Deformed realisation of ABHY Associahedron

In this section, we analyse several classes of deformed realisations of the associahedron in the kinematic space, $\mathcal{K}_n$, of massless particles. We will derive constraints on the deformation parameters such that the canonical form on the deformed realisation generates S-matrix of *some theory* whose field content contains a bi-adjoint massless scalar.

### 4.1  Mixed Scalar Field Amplitude with "fine-tuned" Couplings

We can generalise the analysis of scalar field theory with mass $m$ to the case where all the external particles are massless but all the propagators have a pole at $m^2$. For this purpose, we define the deformed kinematic variables as,

$$
\begin{aligned}
\alpha_{ij} &= 1 \, \forall \, (ij) \\
\kappa_{ij} &= \tilde{X}_{ij} = X_{ij} - m^2 \text{ if } |j - i| \geq 2 \\
\kappa_{i,i+1} &= \kappa_{1n} = 0
\end{aligned}
\tag{4.1}
$$

This is equivalent to choosing the following subgroup $\mathcal{G}$.

$$
\mathcal{G} = \mathbf{1} \times (m^2 \cdot \mathbf{1})
\tag{4.2}
$$

The (deformed) Mandelstam variables then turn out to be,

$$
\tilde{s}_{ii+1} = s_{ii+1} - m^2
\tag{4.3}
$$

$$
\tilde{s}_{ij} = s_{ij} \text{ if } |j - i| > 1
\tag{4.4}
$$

This results in the scattering equations,

$$
\begin{aligned}
&\sum_{j \neq i} \frac{\tilde{s}_{ij}}{\sigma_{ij}} = 0 \\
&\implies \sum_{j \neq i} \frac{p_i \cdot p_j}{\sigma_{ij}} - \frac{1}{2} m^2 \left( \frac{1}{\sigma_{i,i+1}} + \frac{1}{\sigma_{i,i-1}} \right) = 0.
\end{aligned}
\tag{4.5}
$$

As all the external particles are massless, the kinematic space $\mathcal{K}_n$ is co-ordinatized by $X_{ij}$ variables. The convex realisation obtained through,

$$\tilde{s}_{ij} = -c_{ij} \tag{4.6}$$

is simply the (translate of the) ABHY associahedron in the positive quadrant of the kinematic space $\mathcal{K}_n^+$ and once again has facets at $\kappa_{ij} = 0$ or $X_{ij} = m^2$. It is also immediate that the push-forward of the Parke-Taylor form by diffeomorphisms in 4.5 generate the canonical form on the associahedron realised through eqns. 4.6

$$\sum_{\text{solns}} \omega_n = \sum_T \prod_{(kl) \in T} \frac{1}{\kappa_{kl}} \bigwedge_{(ij) \in T_0} \mathrm{d}^{n-3} \kappa_{ij} \tag{4.7}$$

This form equals the (color ordered) tree-level n-point amplitude for a scalar field theory with two fields $\phi_1$, $\phi_2$ with masses $m_1 = 0$, $m_2 = m$ and with a interaction of the type $(\phi_2^3 + \phi_1^2 \phi_2 + \phi_1 \phi_2^2)$ where all the terms have the same coupling. Hence even in this case, the scattering equations $\sum_{j \neq i} \frac{\tilde{s}_{ij}}{\sigma_{ij}} = 0$ generate diffeomorphisms between the worldsheet associahedron and a rigid translate of the ABHY associahedron.

All the positive geometries reviewed so far are such that the set of all the normals to the co-dimension one facets is a global translation of realisation [8, 9]. In section 4.2, we will consider one of the simplest deformations in which the deformed realisation of $A_{n-3}$ is not simply a rigid translation (that is, not all the $\alpha_{ij}$ are equal). As we will see, this results in a positive geometry for the S-matrix in which all the external states are massless and all the propagators are massive and where all the interactions are cubic.

## 4.2 One-parameter deformation

One of the simplest non-trivial deformation maps is parametrized by a single parameter $\alpha$.

$$
\begin{aligned}
m_{ij} &= m \,\forall\, (i,j)\,|\, n-2 \geq |j-i| \geq 2 \\
\alpha_{ij} &= 1 \,\forall\, |i-j| = 2 \,\mathrm{mod}\, n \\
\alpha_{ij} &= \alpha \text{ otherwise}
\end{aligned} \tag{4.8}
$$

We note that for $n < 6$ all the $\alpha_{ij} = 1$ and hence the resulting realisation is simply the ABHY associahedron with all the co-dimension one facets at $X_{ij} = m^2$.

On the other hand, for $n \geq 6$, given a reference triangulation $T_0$, the corresponding realisation obtained using $\tilde{s}_{ij} = -c_{ij} \,\forall\, (i,j) \notin T_0^c$ is not simply a (displaced) ABHY associ-ahedron. Let us first analyse the $n = 6$ case to illustrate this point. Let $T_0 = \{13, 14, 15\}$ so that the associahedron is realised in the hyper-plane co-ordinatized by $X_{13}, X_{14}, X_{15} \geq m^2$. Let $d_{ij} = \frac{c_{ij}}{\alpha_{ij}}$. The realisation that we denote as $A_n^\alpha$ is the closed convex intersection bounded

by the planes,

$$\begin{aligned}
\tilde{X}_{24} &= -\tilde{X}_{13} + \alpha \tilde{X}_{14} + d_{24} \\
\tilde{X}_{35} &= \tilde{X}_{15} - \alpha \tilde{X}_{14} + d_{35} \\
\tilde{X}_{46} + \tilde{X}_{15} &= d_{46} \\
\tilde{X}_{36} + \tilde{X}_{14} &= d_{36} \\
\tilde{X}_{26} + \tilde{X}_{13} &= d_{26} \\
\tilde{X}_{25} &= \tfrac{1}{\alpha}\left(\tilde{X}_{15} - \tilde{X}_{13}\right) + d_{25}
\end{aligned} \tag{4.9}$$

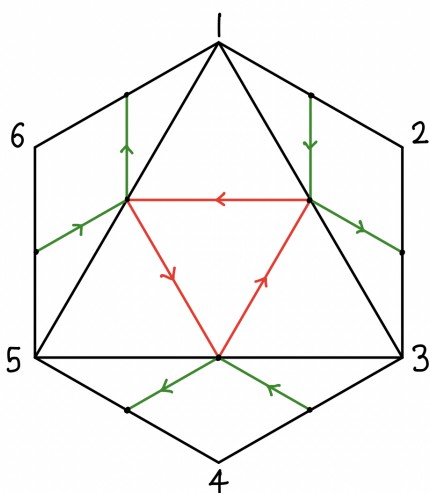

**Figure 2**: Triangulation of hexagon with cyclic quiver.

As before, we can now compute the push-forward of the Parke-Taylor form using the deformed scattering equations. Let $T_c$ be the set of three triangulations whose dissection quiver is cyclic as shown in the figure 2, and let $T_{\mathrm{ac}}$ be the complimentary set. It can now be easily verified that the canonical form equals,

$$\Omega_{n=6}|_{A_3^\alpha} = \sum_{t \in T_c} \frac{\alpha}{\prod_t \tilde{X}_{ij}} + \sum_{t \in T_{\mathrm{ac}}} \frac{1}{\prod_t \tilde{X}_{mn}} \tag{4.10}$$

(Up to an overall scaling) this form equals the colored ordered tree-level amplitude in a theory with the following interaction.

$$V(\phi_1, \phi_2) = \lambda_1\,\phi_1^2\,\phi_2 + \lambda_2\,\phi_1\phi_2^2 + \lambda_3\phi_2^3 \tag{4.11}$$

where $\phi_1$ is massless and $\phi_2$ is massive of mass $m$. The mapping of the form to the amplitude is through the following relation.

$$\alpha = \frac{\lambda_1 \lambda_3}{\lambda_2^2} \tag{4.12}$$

$\Omega_6|_{A_3^\alpha}$ is precisely the push-forward of the Parke-Taylor form that we get using eqns. (3.11, 3.12).

This result can be immediately generalised to arbitrary number of external particles. Consider the $n$-gon with external edges dual to the massless particles. The identification of the $\Omega_n|_{A_{n-3}^\alpha}$ with an amplitude follows from the fact that each triangulation of the $n$-gon is dual to a Feynman diagram with every triangle being dual to a cubic vertex. This implies that given any two triangulations $T_1$, $T_2$ we have the following relation.

$$\prod_{e \in T_1} \alpha_e \prod_{\triangle \in T_1} \lambda_\triangle = \prod_{e \in T_2} \alpha_e \prod_{\triangle \in T_2} \lambda_\triangle \tag{4.13}$$

where $e$ indicates the chords and $\triangle$ are the cells of the triangulation.

Let $\{T_c^I\}$ be the set of all triangulations that have $1 \le I \le [\frac{n}{2}] - 2$ closed cycles in their corresponding quivers. Let $\{T_{ac}\}$ be the complimentary set. The push-forward of the Parke Taylor form evaluated gives the following form on the $A_{n-3}^\alpha$.

$$\Omega_{n-3} = \left[ \sum_{I=1}^{\left[\frac{n}{2}\right]-2} \sum_{T_{cc}^I} \left(\frac{1}{\alpha}\right)^I \prod_{(ij) \in T_{cc}^I} \frac{1}{\tilde{X}_{ij}} + \sum_{T_{ac}} \prod_{(ij) \in T_{ac}} \frac{1}{\tilde{X}_{ij}} \right] \bigwedge_{(mn) \in T_0} d\tilde{X}_{mn} \tag{4.14}$$

Identifying $\alpha$ as $\frac{\lambda_1 \lambda_3}{\lambda_2^2}$ implies that $\Omega_n|_{A_{n-3}^\alpha}$ is the tree-level amplitude for the theory with two scalar interaction in 4.11.

Although this realisation is perhaps one of the simplest possible deformations of the ABHY realisation, we note that from the perspective of the CHY formula, the result is a happy surprise : The S-matrix involving mass-less external states that interact via a massive scalar of mass $m$ can be obtained by integrating the Parke-Taylor form over the moduli space, where the integration is localised over the solution of the deformed scattering equations defined in eqn. 4.5.

## 4.3 Multi-parameter deformations.

Encouraged by the result of the previous section, we consider a more intricate isometry between the embedding space and kinematic space.

$$\begin{aligned} \mathcal{K}_n &\to \mathbb{R}^{\frac{n(n-3)}{2}} \\ X_{ij} &\to \kappa_{ij} := \alpha_{ij} \left(X_{ij} - m_{ij}^2\right) \end{aligned} \tag{4.15}$$

where

$$\begin{aligned} \alpha_{ij} &= \alpha_{kl} \text{ and } m_{ij} = m_{kl} \\ &\text{if and only if } |i - j| = |k - l| \bmod n \end{aligned} \tag{4.16}$$

These transformations clearly form a group $\subset (\mathbf{R}^\star)^{\frac{n(n-3)}{2}} \times \mathbf{R}^{\frac{n(n-3)}{2}}$.

The ABHY realisation in the embedding space is then deformed in $\mathcal{K}_n^+$. The pull-back of the planar scattering form on the deformed realisation is once again the push-forward of the Parke-Taylor form on the resulting convex realisation via deformed scattering equations.

We will now show that the $n - 3$ form on $A_{n-3}^{\{\alpha\}}$ is the $n$ point amplitude of a local and unitary QFT with the Lagrangian

$$L_n(\phi_1, \phi_2, \ldots, \phi_{\lfloor \frac{n}{2} \rfloor}) = \frac{1}{2} \sum_{I=1}^{\lfloor \frac{n}{2} \rfloor} \text{Tr}(\partial_\mu \phi_i \cdot \partial_\mu \phi_i) - \sum_{\substack{1 \leq I \leq J \leq K \leq \lfloor \frac{n}{2} \rfloor \\ I+J=K \bmod n}} \frac{\lambda_{IJK}}{3!} \phi_I \phi_J \phi_K \qquad (4.17)$$

where $\phi_1$ is a massless bi-adjoint scalar and $\phi_I\, I > 1$ are massive scalars with distinct masses $m_I \neq 0$. As we prove, for generic value of the couplings, the canonical form is not the $n$ point amplitude associated to $L_n$, however if the couplings satisfy a set of relations, then the deformed associahedra indeed generate amplitude in a sequence of theories.

Let us first analyse the convex realisation in $\mathcal{K}_n^+$. We choose $T_0 = \{\, 13, \ldots 1n-1 \,\}$. The convex realisation is then obtained by using,

$$\tilde{s}_{ij} = -c_{ij} \,\forall\, (i,j) \notin T_0^c = \{\, 24, \ldots, 2n-1 \,\} \qquad (4.18)$$

As before, the push-forward of the Parke-Taylor form will generate the following top form on the resulting convex realisation.

$$\Omega_{n-3} \big|_{A_{n-3}^{\{\alpha\}}} = \sum_T \frac{\prod_{(i,j) \in T_0} \alpha_{ij}}{\prod_{(m,n) \in T} \alpha_{mn}} \frac{1}{\prod_{(m,n) \in T} \tilde{X}_{ij}} d^{n-3} \tilde{X}_{T_0} \qquad (4.19)$$

Can this form be mapped to S-matrix of a local QFT ? In order to answer this question, we have to map the deformation parameters $\alpha_{ij}$ which are $\lfloor \frac{n}{2} \rfloor - 1$ in number to the space of couplings between $\lfloor \frac{n}{2} \rfloor$ species of scalar fields with masses $m_1 = 0, m_2 = m_{13} \ldots, m_{\lfloor \frac{n}{2} \rfloor} = m_{1 \lfloor \frac{n}{2} \rfloor}$ respectively.

The set of all possible cubic couplings among these fields correspond to all the distinct (colored) triangles in an $n$-gon that can be drawn with $\lfloor \frac{n}{2} \rfloor$ distinct internal chords and an external edge. We will denote the cardinality of this set as $N_c$. An explicit formula for $N_c$ can be found in appendix A.

As an explicit check, we can verify that for $n = 6, 8, 10, 12, 14$ we indeed have, $N_c = \{3, 5, 8, 12, 14\}$ respectively. We note that, in these cases the number of deformation parameters are $\mathcal{D}_c = \{\, 2, 3, 4, 5, 6 \,\}$ respectively.

The number of deformation parameters is thus generically smaller than number of kinematically allowed independent couplings. However, for $n \in \{5, \ldots, 8\}$ this difference is not constraining and the canonical form on the deformed associahedra of dimension $\leq 6$ are the tree-level amplitude with massless external particles and with the interaction Lagrangians respectively given by,

$$\begin{aligned} V_1 &= \lambda_{112}\, \phi_1^2\, \phi_2 + \lambda_{123}\, \phi_1\, \phi_2\, \phi_3 + \lambda_{222}\, \phi_2^3 \\ V_2 &= \lambda_{112}\, \phi_1^2\, \phi_2 + \lambda_{123}\, \phi_1\, \phi_2\, \phi_3 + \lambda_{134}\, \phi_1\, \phi_3 \phi_4 + \lambda_{222}\, \phi_2^3 + \lambda_{224} \phi_2^2 \phi_4 + \lambda_{233}\, \phi_2\, \phi_3^2 \end{aligned} \qquad (4.20)$$

We now give an explicit example of "bootstraping" forms to obtain the amplitude. If $n = 6$, the canonical form is the tree-level planar amplitude if we map the deformation parameters

to the ratio of the couplings that generate the interaction $V_1$ as,

$$\frac{\alpha_{13}}{\alpha_{14}} = \frac{\lambda_{123}^2}{\lambda_{222}\lambda_{112}} \tag{4.21}$$

And similarly in the $n = 8$ case, the mapping from deformation parameters to $V_2$ defined via,

$$\begin{aligned}\frac{\alpha_{13}}{\alpha_{14}} &= \frac{\lambda_{213}\lambda_{314}}{\lambda_{224}\lambda_{112}} \\ \frac{\alpha_{13}}{\alpha_{15}} &= \frac{\lambda_{314}^2}{\lambda_{323}\lambda_{112}}\end{aligned} \tag{4.22}$$

However as we prove below, the connection between positive geometry, canonical form and S-matrix is more subtle if $n \geq 9$. In this case, the canonical form equals the amplitude of a theory only if the couplings are not all independent and the number of relations between the couplings equals the number of $\lambda_{IJK}$ for $2 < I \leq J \leq K$.

Before proving this statement and finding the constraints among couplings for which the canonical form is the S-matrix, we first need to introduce certain definitions.

**Definition 4.1** (Colored dissection quiver). Given a colored triangulation $T_c$, a colored dissection quiver is a dissection quiver [9] associated to $T_c$ in which the vertices are coloured by coloring of the edges.(See figure 3)

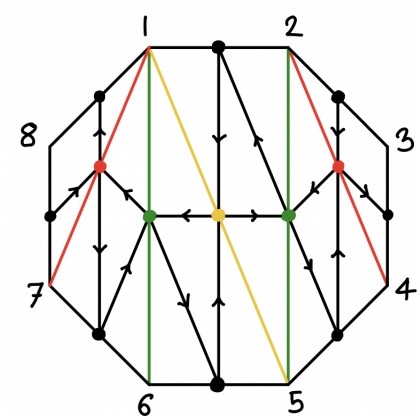

**Figure 3**: Colored dissection quiver.

Given a colored dissection quiver, any cycle of length 3 will be denoted as $c_{IJK}$ where $I$, $J$, $K$ are the colorings associated to the three edges of the cycle.

**Definition 4.2** ( Equivalence class of colored triangulations). $[\mathcal{C}_{(I_1 J_1 K_1) \dots (I_m J_m K_m)}]$ is an equivalence class of colored dissection quivers, where two quivers are considered equivalent if and only if they both have $m$ closed 3 cycles with the set of colorings $I_1 J_1 K_1 \dots I_m J_m K_m$. $[\mathcal{C}_{(I_1 J_1 K_1) \dots (I_m J_m K_m)}]$ defines a unique equivalence class of triangulations $[T_{(I_1 J_1 K_1) \dots (I_m J_m K_m)}]$.

The canonical form on the deformed associahedron is an S-matrix if the deformation parameters are mapped onto the (ratio of) couplings which are subjected to certain constraints derived thanks to the lemmas below.

**Lemma 4.1.** Consider two vertices $v_1$, $v_2$ of $A_{n-3}^{\{\alpha\}}$ that correspond to $T_1, T_2 \in [\, T_{(I_1 J_1 K_1) \dots (I_m J_m K_m)} \,]$, then we have,

$$\text{res}_{v_1} \Omega_{n-3}\big|_{A_{n-3}^{\{\alpha\}}} = \text{res}_{v_2} \Omega_{n-3}\big|_{A_{n-3}^{\{\alpha\}}} \tag{4.23}$$

*Proof.* This statement can be proved simply by showing that for $T_1$ and $T_2$ whose dissection quivers are acyclic,

$$\prod_{e_1 \in T_1} \alpha_{b(e_1)f(e_1)} = \prod_{e_2 \in T_2} \alpha_{b(e_2)f(e_2)} \tag{4.24}$$

Where $e_i$ denote the dissection chords in $T_i$ with $b(e_i)$, $f(e_i)$ being the two end points of $e_i$.[4] Now let $T_1 = \{\, 13, \dots, 1n-1 \,\}$. The LHS of eqn.(4.24) is

$$\prod_{e_1 \in T_1} \alpha_{b(e_1)f(e_1)} = \alpha_{13}^2 \dots \alpha_{1n-1}^2 \tag{4.25}$$

The equality in eqn.(4.24) will be violated if and only if there is atleast one $\alpha_{1i}$ which does not contribute quadratically in the right hand side. However this is only possible if the chord $(1i)$ is mutated to a different chord. As such a chord can not be of the form $(1, j)$ for some $j$, we will necessarily have a closed triangle loop of three chords for all such triangulations $T_2$. This completes the proof. $\qquad \square$

**Lemma 4.2.** The pull back of the planar scattering form on $A_{n-3}^{\{\alpha\}}$ $\Omega_{n-3}\big|_{A_{n-3}^{\{\alpha\}}}$ is a tree-level color ordered S-matrix for an interacting QFT involving $\lfloor \frac{n}{2} \rfloor + 1$ species of bi-adjoint scalars with masses $m_I \,|\, I = 1, \dots, \lfloor \frac{n}{2} \rfloor$ with the following constraints on the couplings.

- All the couplings $\lambda_{IJK}$ are non-zero if and only if $I + J = K$ modulo $n$ and

- The number of relations this set of couplings have to satisfy equals the number $\mathcal{C}_n$ of 3-cycles $c_{IJK}$ in the colored dissection quiver where $I, J, K > 2$ modulo $n$.

We prove this lemma in appendix B. The number $\mathcal{C}_n$ can be computed using the following lemma.

**Lemma 4.3.** The number of relations the set of coupling $\lambda_{IJK}$ have to satisfy is given by the following formula. Let $n = 3\tilde{n} + n_1$ with $n_1 \in \{\, 0, 1, 2 \,\}$.

$$\mathcal{C}_n = \sum_{I=3}^{\tilde{n}} \left( \lfloor \frac{n}{2} \rfloor - \lceil \frac{I}{2} \rceil \right) \tag{4.26}$$

---

[4]For the purpose of this argument, orientation of $e_i$ is not relevant.

*Proof.* We want to compute the cardinality of the set $\{\lambda_{IJK} \mid 3 \leq I \leq J \leq K \leq \lfloor \frac{n}{2} \rfloor \mid I + J = K \text{ modulo } n\}$. It suffices to consider $I \leq J \leq K$ as the coupling $\lambda_{IJK}$ is symmetric in $I$, $J$ and $K$. This means

$$I \in \{3 \ldots, \lfloor \frac{n}{3} \rfloor =: \tilde{n}\} \tag{4.27}$$

For any given $I$, the range of $J$ is

$$J \in \{I, \ldots, \lfloor \frac{n}{2} \rfloor - \lceil \frac{I}{2} \rceil\} \tag{4.28}$$

$J$ can not exceed $J_{\max} = \lfloor \frac{n}{2} \rfloor - \lceil \frac{I}{2} \rceil$ as when $J = J_{\max}$,

$$K = \frac{n}{2} - I + \lceil \frac{I}{2} \rceil = J_{\max} \text{ or } J_{\max} + 1 \tag{4.29}$$

Clearly if $J > J_{\max}$ then $K$ is less than $J$ which is a contradiction. Formula for $\mathcal{C}_n$ is a direct consequence of eqns. (4.27, 4.28). $\qquad \square$

It can be verified immediately that $\mathcal{C}_{10} = 1$, $\mathcal{C}_{12} = 3$, and so on. Hence for $n \geq 9$ the canonical form equals tree-level planar amplitude associated to a local interaction in $N_c$ kinematically allowed couplings are constrained by $\mathcal{C}_n$ relations. And in this case, (up to an overall scaling set by $\prod_{i=1}^{\lfloor \frac{n}{2} \rfloor - 1} \lambda_{1 \, i \, i+1}$ ) the canonical form $\Omega_n|_{A_{n-3}^d}$ equals the n-point color ordered amplitude $M_n(1 \ldots n \mid 1, \ldots, n)$ of a quantum field theory, with the interaction Lagrangian

$$V(\phi_1, \phi_2, \ldots, \phi_{\lfloor \frac{n}{2} \rfloor}) = \sum_{\substack{1 \leq I,J,K \leq \lfloor \frac{n}{2} \rfloor \\ I+J=K \text{ modulo } n}} \frac{1}{3!} \lambda_{IJK} \, \phi_I \, \phi_J \, \phi_K \tag{4.30}$$

Above result thus "bootstraps" a push-forward of the $n-3$ dimensional Parke-Taylor form to an $n$ point tree-level planar S matrix of a multi-scalar bi-adjoint QFT. It is important to note that unlike the canonical form on the undeformed ABHY realisation (which, for all $n$ is the S-matrix of a $\phi^3$ theory with a single bi-adjoint scalar field) the canonical form on deformed realisation and the associated forms generate amplitudes in a family of theories parametrized by $n$.

A classification of the class of S-matrices that result from the more general deformed realisations of the associahedron is beyond the scope of this paper. However in the next section, we ask a reverse question instead. Namely does a S-matrix of a local scalar field Lagrangian with massive as well as massless poles fit into the positive geometry program? As we show in section 5, the answer is in the affirmative for the simplest possible Lagrangian with two (bi-adjoint) scalars.

## 5 From a Lagrangian to a Deformed Realisation

We can rephrase the question stated above as follows : Given the interaction defined in eqn.(4.30), is tree-level color order $m$-point amplitude (when $m \neq n$) a (weighted sum over) canonical forms associated to positive geometries?

Although we do not answer this question in the current paper for interaction defined in eqn.(4.30), it is reminiscent of a question we had investigated in [1]. In [1], it was shown that the perturbative S-matrix for an $n$ point amplitude of mass less particles with interaction of the form

$$V(\phi_1, \phi_2) = \lambda_1 \, \phi_1^3 \, + \, \lambda_2 \phi_1^2 \phi_2 \tag{5.1}$$

is a weighted sum over $n-3$ forms uniquely defined by a class of positive geometries known as colorful associahedron. A colorful associahedron is a combinatorial polytope which is an associahedron in which a specific set of co-dimension one facets is colored distinctly as opposed to the other (uncolored) facets. (we will review the precise definition of the colorful associahedron below.) The final result of the analysis in [1] can be summarised as follows.

Let us denote the $n$ point amplitude up to order $\lambda_2^2$ as $M_n^{(2)}$. It was proved in [1] that $M_n^{(2)}$ is a weighted sum over canonical forms associated to the colorful associahera as well as the ABHY associahedron all of whose boundaries correspond to massless poles. All the weights multiplying forms of the colorful associahedra were positive and the weight associated to the "massless" ABHY associahedron was negative.

In light of the insights we have gained in this paper, we improve on the analysis performed in [1] and show that the $n$ point amplitude $M_n^{(2)}$ is a weighted sum over certain deformed realisation of $A_{n-3}$ (with no contribution of the undeformed ABHY associahedron with all it's boundaries at $X_{ij} = 0$) such that all the weights are positive. [5]

Let $\mathcal{S}_{ij}$ be a subset of chords defined as,

$$\mathcal{S}_{ij} = \{ (i,j), (i+1, j+1), \ldots, (i + |j-i| - 1, j + |j-i| - 1) \} \tag{5.2}$$

Now consider the following map between the embedding space and $\mathcal{K}_n$,

$$\begin{aligned}\kappa_{ij} &= \alpha \, (X_{ij} - m^2) \; \forall \; (i,j) \; \in \mathcal{S}_{ij} \\ &= X_{ij} \, \text{otherwise}\end{aligned} \tag{5.3}$$

Given any reference triangulation $T$ of the $n$-gon, We can now use

$$\bar{s}_{kl} = - c_{kl} \, \forall \, (k,l) \notin T^c \tag{5.4}$$

to obtain *the deformed realisation* $A_{n-3}^{T,(i,j)}$ *of the associahedron* $A_{n-3}$. It is clear that precisely $|j-i|$ number of facets correspond to $X_{ij} = m^2$ pole, whereas rest of the facets correspond

---

[5]Set of all these deformed realisations is in bi-jection with the set of all colorful associahedra. We can thus think of the deformed realisations as set of many realisations of the associahedron or each deformed realisation can be thought of as a specific realisation of a unique colorful associahedron.

to $X_{kl} = 0$. The notation $A_{n-3}^{T,(i,j)}$ makes the dependence of the realisation on $T$ and the distinction between facets associated to massive poles explicit.

Although $A_{n-3}^{T,(i,j)}$ is a realisation of the associahedron $A_{n-3}$, the distinction between two classes of boundaries can also be encoded in the combinatorial definition itself. In other words, we can consider the set of all triangulations in which the chords belonging to the set $\mathcal{S}_{ij}$ are red and the remaining chords are black. In fact, in [1], we had introduced a concept of colorful associahedron that precisely captured this coloring information defined as follows.

**Definition 5.1** (Colorful Associahedron). An $n-3$ dimensional colorful associahedron, $A_{n-3}^{(ij)}$ is an associahedron in which co-dimension one facets corresponding to the chords $(i, j), \ldots, (i + |j - i| - 1, j + |j - i| - 1)$ are red while rest of the facets are black. [6]

As an example, starting with a triangulation $((13)_R, 14)$ of a pentagon, the colorful associahedron $A_2^{(13)}$ has two red edges $(13)_R$, $(24)_R$ while rest of the edges are black.

In the $n - 5$ example, there are in fact 5 possible colorful 2 dimensional associahedra

$$A_2^{(ij)} \,|\, (ij) \,\in\, \{\,(13), (24), (35), (14), (25)\,\} \tag{5.5}$$

with two red edges in each of them. (See figure 4).

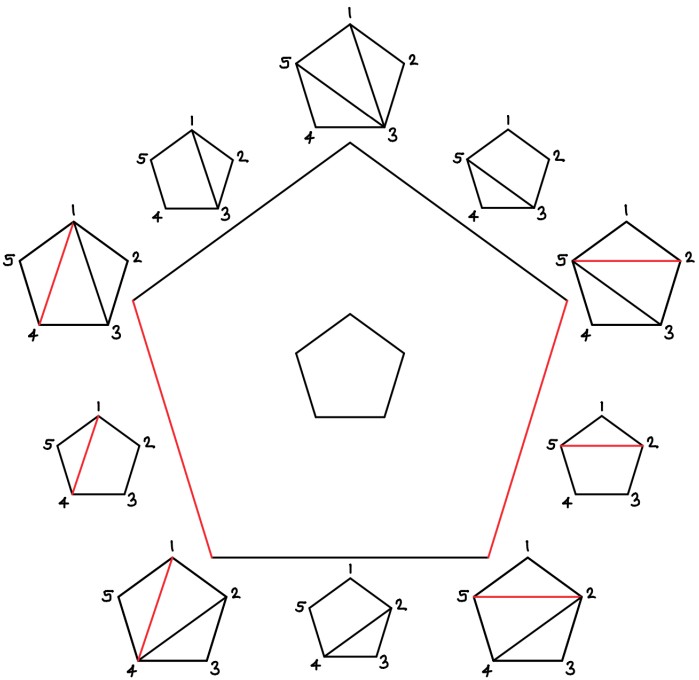

**Figure 4**: Five-point associahedron with $((14)_R, (25)_R)$ faces red.

---

[6]It is "morally" closer to an accordiohedron than the associahedron in the sense that it depends on the reference (partial) triangulation $(ij)$.

Red facets $= \{\, ((13)_R, (24)_R\,), ((24)_R, (35)_R), ((35)_R, (14)_R\,), (\,(14)_R, (25)_R\,), ((25)_R, (13)_R)\,\}$ (5.6)

$A_{n-3}^{T,(i,j)}$ is then the ABHY realisation of the colorful associahedron in the embedding space $\mathbb{R}^{\frac{n(n-3)}{2}}$ which is diffeomorphic to the kinematic space through eqn.(5.3).[7]

We can also compute the push-forward of the Parke-Taylor form using deformed scattering equations. As in the case of massless bi-adjoint theory, this form can also be understood as the planar scattering form on the embedding space, $\Omega_{n-3}$

$$\Omega_{n-3} = \sum_T (-1)^{\sigma(T)} \bigwedge_{(ij)\in T} d\ln \kappa_{ij} \tag{5.7}$$

For $n = 5$,

$$\Omega_2\big|_{A_2^{T_0,(1,3)}} = \alpha \left[\left(\frac{1}{\tilde{X}_{13}\,X_{35}} + \frac{1}{\tilde{X}_{13}\,X_{14}} + \frac{1}{\tilde{X}_{24}X_{14}} + \frac{1}{\tilde{X}_{24}X_{25}}\right) + \frac{1}{X_{35}X_{25}}\right] d\tilde{X}_{13}\wedge dX_{35} \tag{5.8}$$

We can similarly compute $\Omega_2\big|_{A_2^{T,(i,j)}}$ and immediately see that if $\alpha$ is written as $\frac{\lambda_2^2}{\lambda_1^2}$, then

$$\frac{1}{2}\sum_{(ij)\in \mathcal{S}_2}\Omega_2\big|_{A_2^{(ij)_R}} = M_5^{\text{co}}(1,\,\ldots,\,5) \tag{5.9}$$

Rather remarkably, this is a specific example of a general result about color-ordered amplitude as a weighted sum over canonical forms associated to the deformed realisations $A_{n-3}^{T,(ij)}$. For any triangulation $T_0$, we have the following result.

**Lemma 5.1.** Let $\omega_{\lambda_1,\lambda_2}$ be a weighted sum over $\Omega_{n-3}$ defined as,

$$\omega_{\lambda_1,\lambda_2} = \sum_{I=2}^{\lfloor\frac{n}{2}\rfloor}\frac{1}{I}\sum_{(i,j)\mid (j-i)=I}\Omega_n\big|_{A_{n-3}^{T_0,(ij)}} \tag{5.10}$$

Then

$$\omega_{\lambda_1,\lambda_2} = M_n^{\text{co}}\bigwedge_{ij\in T_0}d\tilde{X}_{ij} \tag{5.11}$$

where $M_n^{co}$ is the color-order tree-level amplitude of massless particles with interaction defined in eqn.(5.1)

---

[7]In [1], we had denoted the ABHY realisation of the colorful associahedron as $A_{n-3}^{\mathcal{F}_{ij}}$, where $\mathcal{F}_{ij}$ is the set of all dissections in which precisely $|j-i|$ chords are red. However the information about reference $T_0$ was implicit in this notation.

Proof of this statement directly follows from the proof of lemma (4.1) in [1] where it was shown that, if the colorful associahedron is realised directly in $\mathcal{K}_n^+$, then

$$\omega_{\lambda_1,\lambda_2} = \sum_{I=2}^{\lfloor \frac{n}{2} \rfloor} \frac{1}{I} \sum_{(i,j)\,|\,(j-i)\,=\,I} \Omega_n\big|_{A_{n-3}^{T_0,(ij)}} - \gamma\,\Omega_{n-3}^{m=0}\big|_{A_{n-3}} \tag{5.12}$$

The difference between the two formula simply arises from the fact that in [1], the embedding space for colorful associahedra was identified with the kinematic space. Hence all the vertices (those adjacent to one red facet as well as those adjacent to all uncolored facets) of a given $A_{n-3}^{(i,j)}$ contributed with unit residue. As a result of this, the weighted sum that reproduced the $n$ point amplitude included an extra negative contribution proportional to $n$ point mass-less amplitude in bi-adjoint scalar theory with only $\phi_1$ field.

As we see, the deformed realisation of colorful associahedron incorporates the differential contribution of the two types of vertices and hence a weighted sum *only* over the colorful associahedra is the n-point amplitude of massless external particles and upto order $\lambda_2^2$.

## 5.1 CHY formula for an EFT amplitude

Given a colorful associahedron $A_{n-3}^{T,(i,j)}$ in $\mathcal{K}_n^+$ we can analyse the region where $X_{kl} << m^2\,|\,(k,l) \in \{(i,j),\,\ldots,\,(i+|j-i|-1,j+|j-i|+1)$. This is the limit in which the kinematics of the external massless particles is such that all the diagrams containing $\frac{1}{X_{ij}}$ propagators collapse to a quartic coupling. The perturbative scattering amplitude $O(\lambda_2^2)$ reduces to the amplitude in the low energy effective field theory with a lagrangian,

$$V_{\text{EFT}}(\phi_1) = \lambda_1\phi_1^3 + g\,\phi_1^4 \tag{5.13}$$

where $g = \frac{\lambda_2^2}{m^2}$. $g$ remains finite in the limit $\lambda_2, m \to \infty$ with $\frac{\lambda_2}{m} = $ finite.

It was shown in [1] that the positive geometry for the massless bi-adjoint theory whose canonical form generates S-matrix for eqn.(5.13) is in fact an accordiohedron which is
(1) obtained from dissection containing one quadrilateral and $n-5$ triangles and
(2) is a co-dimension one facet of the colorful associahedron.

"Viewing" the colorful associahedron from $X_{kl} << m^2$ region is equivalent to taking the projection along the $X_{kl} = 0$ facet. In [1], the resulting polytope was referred to as a projected accordiohedron $\mathcal{PAC}[\mathcal{D}_{kl}]$.

**Definition 5.2** (Projected Accordiohedron). Given a colorful associahedron $A_{n-3}^{(ij)}$, let $(k\ell) \in \mathcal{S}_{ij}$. Then a projected accordiohedron, $\mathcal{PAC}[\mathcal{D}(kl)]$ is an accordiohedron generated by the following reference dissection.

$$\mathcal{D}(k\ell) = \{(k, k-2),\, \ldots,\, (k, \ell+1),\, (k+2, \ell),\, \ldots,\, (\ell-2, \ell)\} \tag{5.14}$$

where each vertex pair $(mn)$ is ordered as $m < n$ and all the entries are considered modulo $n$. As an example for $n = 6$ $\mathcal{D}(14)$ is

$$\mathcal{D}(14) = \{(15), (26)\} \tag{5.15}$$

In general an accordiohedron defined using mixed dissections is not a boundary of an associahedron, but a projected accordiohedron by it's very construction is always a co-dimension one face of the colorful associahedron. We can immediately obtain the canonical form on $\mathcal{PAC}(\mathcal{D}(kl))$ by the residue formula,

$$\Omega_{n-4}^{\mathcal{D}(ij)}|_{\mathcal{PAC}(\mathcal{D}(ij))} := \text{Res}_{\kappa_{ij}=0} \, \Omega_{n-3}|_{A_{n-3}^{T,(ij)}} \tag{5.16}$$

We proved in [1] that a weighted sum over $\Omega_{n-4}^{\mathcal{D}(ij)}$ is the tree-level S matrix in a massless bi-adjoint theory with mixed coupling, $\lambda \phi^3 + g \phi^4$, and upto order $g^2$.

$$\Omega_{\text{EFT}} = \sum_{(kl) \in \mathcal{S}_{ij}} a_{(kl)} \, \Omega_{n-4}^{\mathcal{D}(kl)}|_{\mathcal{PAC}[\mathcal{D}(kl)]} \tag{5.17}$$

The weights were explicitly computed in [1] and we refer the reader to the relevant section in that paper for more details.

As we now show, $\mathcal{PAC}[\mathcal{D}(kl)]$ can also be understood as diffeomorphic image of certain co-dimension one boundary of $\overline{\mathcal{M}}_{0,n}(\mathbb{R})$ via (deformed) scattering equations restricted to that boundary. The canonical form on the projected accordiohedron is hence the push-forward of the restriction of the Parke-Taylor form as first conjectured in [13].

Recall that $\mathcal{PAC}[\mathcal{D}(ij)]$ is a boundary of the colorful associahedron defined by $\kappa_{ij} = 0$ for one of the "red" chords. As an example if we consider the colorful associahedron obtained from the reference triangulation $\{(13)_{\text{R}}, \dots\}$. As there are two red facets in this associahedron, there are two possible projected accordiohedra $\mathcal{PAC}[\mathcal{D}(kl)] |(ij) \in \{(13), (24)\}$. Let us consider the projected accordiohedron $\mathcal{PAC}[\mathcal{D}(13)]$. It is immediately cleat that this projected accordiohedron is diffeomorphic image of the $u_{13} = 0$ boundary of the compactified moduli space $\overline{\mathcal{M}}_{0,n}(\mathbb{R})$ under the (deformed) scattering equations.

As the canonical form on $\mathcal{PAC}[\mathcal{D}(ij)]$ is simply the restriction of $\Omega_{n-3}|_{A_{n-3}^{T,(ij)}}$, we see that this form is simply the pushforward of the residue of the Parke-Taylor form on $u_{ij} = 0$ boundary. Hence the tree-level amplitude of 5.1 interaction to order $g$ can be written as a worldsheet formula,

$$\Omega_{\text{EFT}} = \\ \sum_{(kl) \in \mathcal{S}_{ij}} a_{(kl)} \, \int_{\partial_{\sigma_2 = 0} \overline{\mathbf{M}}_{0,n}(\mathbb{R})} \text{Res}_{\sigma_{13} = 0} \, \omega_{ws}(\sigma_3, \dots, \sigma_{n-2}) \\ \prod'_j \, \delta(\sum_{m \neq j} \kappa_{mj} - \phi_{mj}(\sigma_2 = 0, \sigma_3, \dots, \sigma_{n-2})) \tag{5.18}$$

This rather simple corollary of our analysis shows that the perturbative $n$-point amplitude for $\lambda_1 \phi^3 + g\phi^4$ (where $\phi$ is a massless bi-adjoint scalar) is best understood not as a integration over $n-3$ form over the CHY moduli space, but as an integral of lower forms on the boundary of the Moduli space.

## 6 Deformed realisations of one-loop Cluster Polytopes

### 6.1 Review of $D_n$ and $\hat{D}_n$ Cluster Polytopes

The deformed realisation $A_{n-3}^{T,(ij)}$ of an associahedron is a positive geometry for the two-scalar field theory at order $\lambda_2^2$. This realisation can be thought of as the result of a $\mathcal{G}$ action (for certain subgroup $\mathcal{G} \subset (\mathbb{R}^\star)^{\frac{n(n-3)}{2}} \times \mathbb{R}^{\frac{n(n-3)}{2}}$) on the ABHY associahedron. In this section, we argue that this idea (that the deformed realisations $A_{n-3}^{T,(ij)}$ are positive geometries for the S-matrix) extend rather directly to the one loop S-matrix integrands in the same theory.

In [3], Arkani-Hamed, He, Salvatori and Thomas (AHST) introduced $\mathcal{K}_n^{\text{1-L}}$, a planar kinematic space for one loop amplitudes in massless bi-adjoint scalar field theories. It was explained to us by Nima Arkani-Hamed that in fact a careful treatment of tadpole channels require a certain enlargement of the AHST Kinematic space, [4]. Throughout this section, $\mathcal{K}_n^{\text{1-L}}$ denotes this enlarged kinematic space.[8]

The planar kinematic space $\mathcal{K}_n^{\text{1-L}}$ for the 1-loop S-matrix where external momenta satisfy momentum conservation is rather subtle to define. The seminal AHST construction of the kinematic space as well as it's enlargement that we call $\mathcal{K}_n^{\text{1-L}}$ rely on "doubling" the space of external momenta $p_1, \ldots, p_n$ to include the "auxiliary" momentum variables $\bar{p}_1, \ldots, \bar{p}_n$ such that

$$\begin{aligned}
\sum_{i=1}^n p_i + \sum_{i=1}^n \bar{p}_i &= 0 \\
p_i \cdot p_{\bar{j}} &= p_{\bar{i}} \cdot p_j \\
p_i \cdot p_j &= p_{\bar{i}} \cdot p_{\bar{j}}
\end{aligned} \tag{6.1}$$

Subjected to the above constraints, A set of planar kinematic variables can be defined as,

$$\begin{aligned}
X_{ij} &= (p_i + \ldots + p_{j-1})^2 \\
X_{i\bar{j}} &= (p_i + \ldots + p_{\bar{j}-1})^2 \,\forall\, \{\, 1 \leq i < \bar{j} \leq \overline{n-1} \,\}
\end{aligned} \tag{6.2}$$

We can pictorially represent all of the above variables as chords dissecting an abstract $2n$-gon with a hole at the center. Vertices of this polygon are ordered clockwise as $\{\, 1, \ldots, n, \bar{1}, \ldots, \bar{n} \,\}$ (see figure 5).

Except $X_{i\bar{i}} \,|\, 1 \leq i \leq n$, all the remaining chords are straight. As

$$X_{i\bar{i}} = (p_1 + \ldots p_n + p_{\bar{1}} + \ldots p_{\overline{i-1}})^2 \tag{6.3}$$

These "curved" chords precisely correspond to the propagator in a tadpole graph as shown in figure 6.

However these variables do not take into account the Mandelstam invariants which include the loop momentum $l$. These are $\{\, l^2, p_i \cdot l, p_{\bar{i}} \cdot l \,\forall\, i \,\}$. In ( [3,4]) these kinematic invariants were accounted for by using the following variables,

$$\begin{aligned}
Y_i &= l_i^2 \\
\tilde{Y}_i &= l_{\bar{i}}^2
\end{aligned} \tag{6.4}$$

---

[8] We are indebted to Nima Arkani-Hamed for introducing the enlarged Kinematic space and the $\hat{D}_n$ polytope prior to the publication.

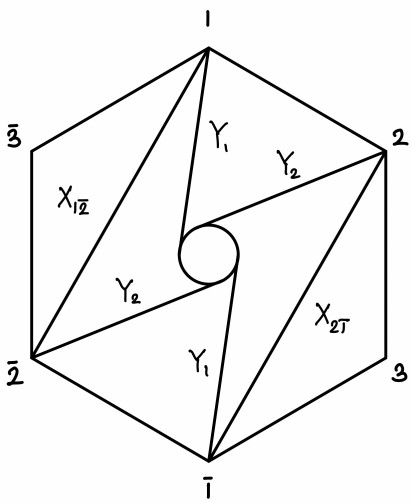

**Figure 5**: Chords in $2n$-gon with a hole at the center.

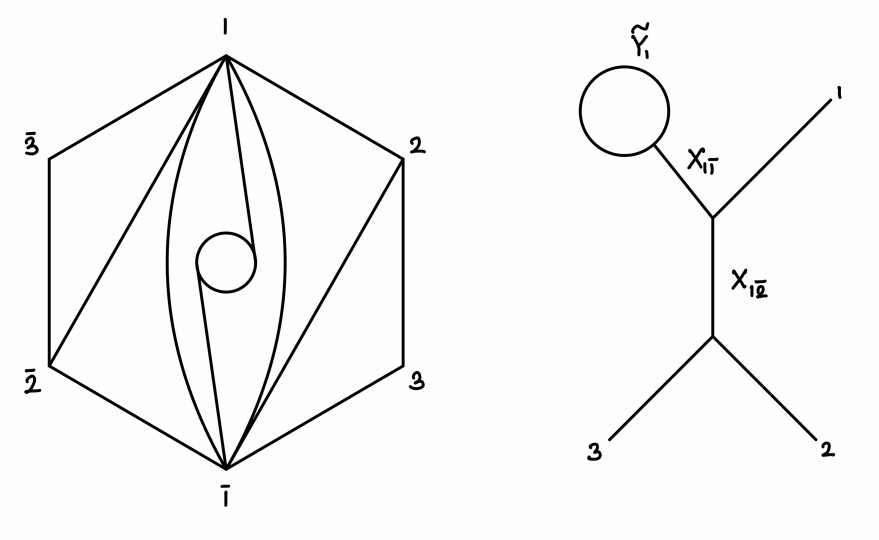

**Figure 6**: Dissections and Feynman diagrams.

where,

$$l_i = (\, l + \sum_{k=1}^{i-1} p_k \,)$$
$$l_{\bar{i}} = (\, l + \sum_{k=1}^{n} p_k + \sum_{m=1}^{i-1} p_{\overline{m}} \,) \tag{6.5}$$

The identification of $Y_1$ with $l^2$ fixes a trivial redundancy in the labelling of the loop momenta. $\mathcal{K}_n^{1-\text{L}}$ is an $n(n+1)$ dimensional space which is co-ordinatized by Four types of variables. We

collectively denote them as $\{X_{IJ}\}$ where the capitalized index $I \in \{1 \ldots, n, 0, \bar{1}, \ldots, \bar{n}\}$. Here 0 labels the annulus in the center of the 2n-gon. We define the following conventions for our labellings :

- The variable $Y_i$ is assigned to a chord which begins at $i($ and $\bar{i})$ and ends on the left (right) side of the annulus.

- Similarly, $\tilde{Y}_i$ begins in $i$ (as well as in $\bar{i}$) and ends on the right (respectively left) side of the annulus.

Geometrically, the set of chords $\{Y_i, Y_{\bar{i}}\}$ can be drawn inside the punctured 2n-gon as in figures 5 and 6.

Thanks to the constraint eqn.(6.1), $X_{IJ}$ are independent planar variables such that all the Mandelstam invariants are linear combinations of $X_{IJ}$.

We denote the set of all chords $(I, J)$ as $\mathcal{PC}_n$.

$$\mathcal{PC}_n = \frac{\{(i,j),\, (i,\bar{j}),\, (i,0),\, (\bar{i},0)\,|\, 1 \le i \le n < \bar{1} \le \bar{i} \le \bar{n}\}}{(i,0) \cup (\bar{i},0) \sim (i,\bar{i})} \tag{6.6}$$

We note that in $\mathcal{PC}_n$, the union of $(i,0)$, $(\bar{i},0)$ is identified with $(i,\bar{i})$ via homotopy equivalence.

Now we consider the following the set of chords.

$$\mathcal{PC}'_n = \{(i,j),\, (i,\bar{j})|_{j \ne i},\, (i,0),\, (\bar{i},0)\,|\, 1 \le i \le n < \bar{1} \le \bar{i} \le \bar{n}\} \tag{6.7}$$

As shown in [16], $\mathcal{PC}'_n$ models quivers of type $D$. Moreover these pseudo-triangulations generate an n-dimensional combinatorial polytope known as $D_n$ whose co-dimension $k$-facets are in 1-1 correspondence with $k$-partial pseudo triangulations generated by all the chords in $\mathcal{PC}'_n$.

It was shown in [3] that there exist convex realisations of $D_n$ polytopes in $\mathcal{K}_n^{1-L}$ which are positive geometries for the 1-loop S matrix for massless bi-adjoint $\phi^3$ theory. A wider class of realisations, all which are positive geometries for the 1-loop bi-adjoint theory were discovered in [17].

However, if instead of considering $\mathcal{PC}'_n$, we consider the set $\mathcal{PC}_n$, then, as shown by Arkani-Hamed, Frost, Plamondon, Salvatori and Thomas in [4], the set of all pseudo-triangulations of the 2n-gon by chords in $\mathcal{PC}_n$ also form a closed convex polytope called the $\hat{D}_n$ polytope.

Every n-point, 1 loop planar Feynman graph with cubic vertices labels two distinct vertices of $\hat{D}_n$ polytope. The distinction between $D_2$ and $\hat{D}_2$ polytopes is shown in figure 7.

We now review the convex realisation of $\hat{D}_n$ polytope inspired by the construction of $D_n$ polytope in [3].[9] Consider the $n(n+1)$ dimensional embedding space $\mathbb{R}^{n(n+1)}$ with the co-ordinates $\{\kappa_{IJ}\}$ labelled by dissections $(I, J) \in \mathcal{PC}_n$.[10]

---

[9]Strictly speaking, AHST proved that the set of constraints that given a reference pseudo-triangulation $PT$ with the corresponding set of kinematic variables $X_{IJ}\,|\,(I, J) \in PT$, all the other $X_{MN}$ are linear combination of $X_{IJ}|(I, J) \in PT$ where the co-efficients in the linear combination are from $\{0, \pm 1\}$ as determined by the $g$ vectors. We assume that this construction goes through verbatim for $\hat{D}_n$ polytope.

[10]we remind the reader, that strictly speaking the embedding space is the real projective space $\mathbb{RP}^{n(n+1)}$ with $\kappa_{ij}$ being the homogeneous co-ordinates.

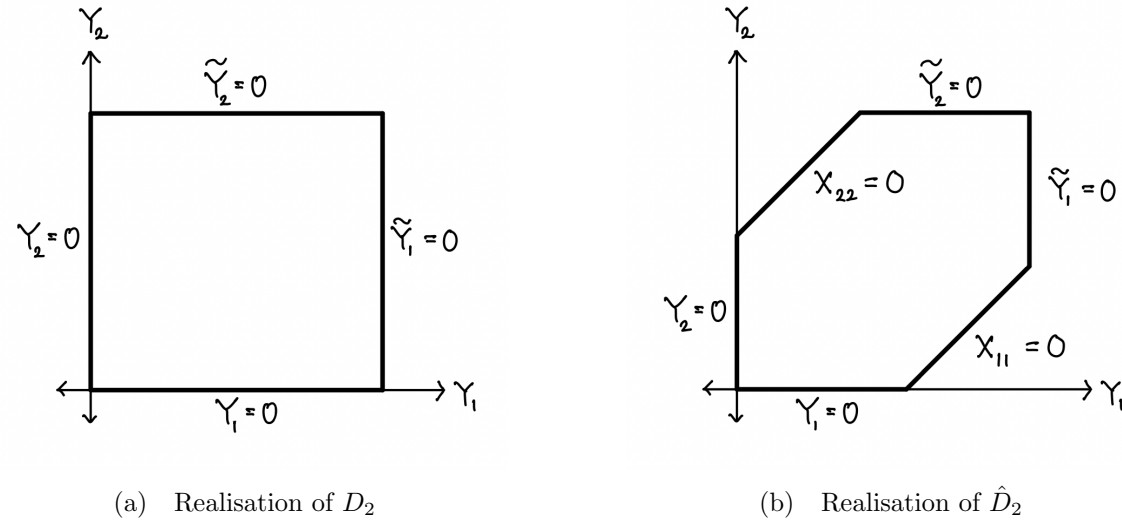

(a)   Realisation of $D_2$                    (b)   Realisation of $\hat{D}_2$

**Figure 7**: Comparison of $D_2$ and $\hat{D}_2$

Consider the "1-loop Mandelstam invariants" in terms of planar kinematic variables.

$$
\begin{aligned}
2p_i \cdot p_j &= \kappa_{i,j+1} + \kappa_{i+1,j} - \kappa_{ij} - \kappa_{i+1,j+1} \\
2p_i \cdot p_{\bar{j}} &= \kappa_{i,\overline{j+1}} + \kappa_{i+1,\bar{j}} - \kappa_{i\bar{j}} - \kappa_{i+1,\overline{j+1}} \\
2l_i \cdot l_{\bar{j}} &= Y_i + \tilde{Y}_j - X_{i\bar{j}}
\end{aligned}
\tag{6.8}
$$

In particular we will denote $2p_i \cdot p_j$, $2p_i \cdot p_{\bar{j}}$, $2l_i \cdot l_{\bar{i}}$ as $s_{ij}^{(1)}$, $s_{i\bar{j}}^{(1)}$, $s_{i0}^{(1)}$ (or equivalently $s_{\bar{i}0}^{(1)}$) respectively.

Let $PT$ be a pseudo-triangulation of the 2n-gon. The AHST realisation of the $D_n$ polytope [3,17] motivates us to seek a realisation of the $\hat{D}_n$ via following set of constraints.

$$
s_{IJ}^{(1)} = -d_{IJ} \forall (I,J) \notin \mathrm{PT}^c
\tag{6.9}
$$

In the $n = 2, 3$ case it can be readily checked that these constraints indeed generate a convex realisation of $\hat{D}_2$, $\hat{D}_3$ inside the positive hyper-quadrant of the embedding space. We will assume that this statement continues to be true for all $n$.

We note that if the embedding space is trivially identified with the $(\mathcal{K}_n^{1-L})$ through,

$$
\kappa_{IJ} = X_{IJ}
\tag{6.10}
$$

then we convex realisations discovered in [3] are inside the positive hyper-quadrant of $\mathcal{K}_n^{1-L}$.

## 6.2   Deformed Realisations of the $D_n$ Cluster Polytope

As in the case of ABHY associahedron, we now study non-trivial isometries between $\mathbb{R}^{n(n+1)}$ to $\mathcal{K}_n^{1-L}$. In particular we discover two deformed realisations $\hat{D}_n^{\{\alpha\}}$ of $\hat{D}_n$ in the kinematic space which extend the results of sections (4.2, 5) to the one loop case respectively.

Consider the following Lagrangian involving two (bi-adjoint) scalar fields.

$$L'(\phi_1, \phi_2) = \sum_{I=1}^{2} \partial_\mu \phi_I \cdot \partial^\mu \phi_I - \frac{1}{2} m^2 \phi_2^2 - (\lambda_1 \phi_1^3 + \lambda_2 \phi_1^2 \phi_2 + \lambda_3 \phi_1 \phi_2^2 + \lambda_4 \phi_2^3) \qquad (6.11)$$

In sections 6.3, 6.4, we prove the following.

- There exists a deformed realisation $\hat{D}_n^{\{\alpha\}}$ which is a positive geometry for the (integrand of the) 1-loop S-matrix involving all massless external states and all massive propagators.

- There exists a family deformed realisation that generalise the construction of colorful associahedron in section 5 such that a weighted sum over all of these realisations define the 1-loop $\phi^3$ integrand with at most one massive propagator.

We have chosen the masses to be $m_1 = 0, m_2 = m$ for simplicity. One can extend our results to the case where the mass of the external states is $M \neq m$.

## 6.3 One parameter deformation

Consider the following map from $\mathcal{K}_n^{1-L}$ to $\mathbf{R}^{n(n+1)}$. It will be convenient for us to label the barred-vertices of the $2n$-gon as $\{n+1, \ldots, 2n\}$.

$$\kappa_{ij} = \alpha_{ij}(X_{ij} - m_{ij}^2) \qquad (6.12)$$

where $m_{ij} = m \,\forall\, (i,j)\,|j-i| \geq 2$. We choose the following ansatz for the deformation parameters.

$$\alpha_{IJ} = \alpha_{KL} \text{ if } |I - J| = |K - L| \qquad (6.13)$$

where $I \in \{1, \ldots, 2n, 0\}$.

**Lemma 6.1.** There exists a choice of the deformation parameters for which the pull back of the canonical form $\Omega_n^{1-L}$ on the deformed realisation $\hat{D}_n^{\{\alpha\}}$ is the 1-loop integrand for the Lagrangian in eqn.(6.11).

*Proof.* Any deformed realisation $\hat{D}_n^{\{\alpha\}}$ is obtained from the convex realisation in the embedding space via a specific $\mathcal{G}$ action. The AHST realisation in the embedding space is defined by using a reference pseudo-triangulation that we denote as $\mathrm{PT}_0$. For convenience we choose it as

$$\mathrm{PT}_0 = \{(1,0), (3,0)\,(1,3), (3,\bar{1}), \ldots\} \qquad (6.14)$$

where ... denote *any* set of straight chords that complete the pseudo-triangulation of the $2n$-gon starting from $\{(1,0), (3,0), (1,3), (3,\bar{1})\}$.

It can be readily checked that if we denote the pull-back of $\Omega_n^{1-\mathrm{L}}$ on $\hat{D}_n^{\{\alpha\}}$ by $\omega_n^{\{\alpha\}}$, then

$$\omega_n^{\{\alpha\}} = \sum_{PT} \Big[ \frac{\prod_{(K,L) \in PT_0} \alpha_{KL}}{\prod_{(M,N) \in PT} \alpha_{MN}} \prod_{(M,N) \in PT} \frac{1}{\tilde{X}_{MN}} \Big] \wedge_{(P,Q) \in PT_0} d\tilde{X}_{PQ} \tag{6.15}$$

Hence, the pull-back equals the 1-loop integrand (with all massive propagators) if and only if for any pseudo-triangulation $PT_1$ which is dual to a 1-loop Feynman graph $PT_1^\star$, deformation parameters satisfy the following set of constraints.

$$\prod_{(I,J) \in PT_0} \alpha_{IJ} \prod_{t \in PT_0} \lambda_t = \prod_{(K,L) \in PT_1} \alpha_{KL} \prod_{t_1 \in PT_1} \lambda_{t_1} \tag{6.16}$$

If for all the straight chords $(i, j)$ we choose the deformation parameters as

$$\alpha_{ij} = \alpha = \frac{\lambda_3^2}{\lambda_2 \lambda_4} \ \forall \ (i, j) \ \in \{1, \ldots, 2n\}, \ |j - i| = 2 \tag{6.17}$$

$$\alpha_{ij} = 1 \forall |j - i| \geq 3 \tag{6.18}$$

then all the constraints in eqn.(6.16) are satisfied if $PT_1$ differs from $PT_0$ by mutations which only mutate the straight chords.

If $PT_1$ differs from $PT_0$ by a single mutation $(i, i+2) \rightarrow (i+1, 0)$ (assuming that this mutation is permissible from $PT_0$) then

$$\alpha_{i,0} = 1 \tag{6.19}$$

ensures that

$$\alpha_{i,i+2} \, \lambda_2^2 \, \lambda_4^2 = \alpha_{i+1,0} \, \lambda_2 \lambda_4 \, \lambda_3^2 \tag{6.20}$$

is satisfied. Finally we note that a mutation from $(1, 0)$ or $(3, 0)$ to $(3, \overline{3})$ or $(1, \overline{1})$ does not change the overall coupling in a Feynman graph with all massive propagators and hence we can choose

$$\alpha_{i,\overline{i}} = 1 \tag{6.21}$$

Eqns (6.17, 6.19 and 6.21) show that there exists one-parameter deformation (parameter being $\alpha = \frac{\lambda_3^2}{\lambda_2 \lambda_4}$ ) of $\hat{D}_n$ which (1) satisfies the chosen ansatz in eqn.(6.13) and (2) produce $\omega_n^{\{\alpha\}}$ which is the desired 1-loop integrand for an S-matrix involving all massless external states and all massive propagators. $\qquad\square$

Lemma 6.1 in conjunction with the result in section 4.2 shows that there exists a deformed realisation of both, ABHY associahedron as well as AHST $\hat{D}_n$ which are positive geometries of the tree-level and one-loop S-matrix in which all the external particles are massless and all the propagators are massive.

## 6.4 Multi-parameter deformation

In the previous section we discovered a deformed realisation of $\hat{D}_n$ polytope whose boundaries correspond to massive poles of the 1-loop S matrix integrand when all the external states are massless. The complete 1-loop integrand for eqn.(6.11) with massless external states can be regrouped into terms classified according to number of massive propagators in each Feynman graph. As we have seen, the sum over all the diagrams in which all propagators are massless is the (pull-back) of the canonical form on AHST realisation of the $\hat{D}_n$ polytope and the sum over all the diagrams in which all the propagators are massive is the deformed realisation obtained in the previous section.

We now show that the sum over all the diagrams in which at most one propagator is massive is a weighted sum over a specific family of deformed realisations. This result extend the tree-level results in section (5) to the one-loop case.

As we will see, the relevant deformed realisations of $\hat{D}_n$ polytope mirror the construction of the colorful associahedron $A_{n-3}^{T,(ij)}$ in section 5

We start by introducing some objects which are necessary to define the deformation map which will extend the results of section 5 to one loop.

Given any chord, $(I, J) \in \mathcal{PC}_n$, we define a set of chords $\mathcal{S}_{IJ}$ which

$$(1) \text{ Intersect } (I, J) \text{ in the interior of the polygon and}$$
$$(2) \text{ All of which have "length" equal to } |I - J|.$$

We refer to the chord $(I, J)$ as the seed of $\mathcal{S}_{IJ}$. Seeds come in Four types. $\{(i, j), (i, \bar{j}), (i, 0), (i, \bar{i})\}$. As we will see, the seed $(\bar{i}, 0)$ is equivalent to $(i, 0)$ in the sense that they both generate the same $\mathcal{S}$ set defined above.

If the seed is a straight chord $(i, j)$ (or $(i, \bar{j})$) with $1 \le i \le n$ and $n - 1 \ge |j - i| \ge 2$, then we have

$$\mathcal{S}_{ij} = \{(i, j), \ldots, (i + |j - i| - 1, j + |j - i| - 1)\} \tag{6.22}$$

The restriction over the range of $i$ is simply to avoid redundancy. The co-dimension one facets of $\hat{D}_n$ that are in 1-1 correspondence with 1-partial pseudo-triangulations $(\bar{i}, \bar{j})$ correspond to the same physical pole in the kinematic space as those facets which are in bijection with $(i, j)$.

For the remaining types of seeds, $\mathcal{S}$ sets are,

$$\begin{aligned} \mathcal{S}_{i\bar{i}} &= \{X_{1\bar{1}}, \ldots, X_{n\bar{n}}\} \forall i \\ \mathcal{S}_{i,0} &= \{Y_i, \tilde{Y}_i,\} \forall 1 \le i \le n \end{aligned} \tag{6.23}$$

We now define choose the following (multi-parameter) deformation.

Let $(M, N)$ be a seed not of the type $(i, 0)$.

$$\begin{aligned} \kappa_{IJ}^{(MN)} &= \alpha_{MN} (X_{IJ} - m^2) \; \forall \; (I, J) \in \mathcal{S}_{MN} \\ &= X_{IJ} \text{ otherwise} \end{aligned} \tag{6.24}$$

If $(M, N) = (i, 0)$ for some $i$, then

$$
\begin{aligned}
\kappa_{I,J}^{(i,0)} &= \alpha\,(\,X_{I,0} - m^2\,) \text{ if } (\,I \in \{i, \bar{i}\}, \ J = 0\,) \\
&= \beta\,X_{i\bar{i}} \text{ if } (I, J) = (i, \bar{i}) \\
&= X_{IJ} \text{ otherwise}
\end{aligned}
\tag{6.25}
$$

Given a seed and any (positive) choice of the deformation parameters, we have a deformed realisation of $\hat{D}_n$ polytope.[11] We illustrate these ideas with a few simple examples.

- Consider $n = 2$. In this case $\hat{D}_2$ is a two dimensional polytope whose boundaries span a hexagon. We can list all the $\mathcal{S}_{IJ}$ as follows.

$$
\begin{aligned}
\mathcal{S}_{1,0} &= \{Y_1,\ \tilde{Y}_1\} \\
\mathcal{S}_{2,0} &= \{Y_2\,\tilde{Y}_2\} \\
\mathcal{S}_{1,\bar{1}} &= \{X_{1\bar{1}},\ X_{2\bar{2}}\}
\end{aligned}
\tag{6.26}
$$

Hence we have three deformed realisations of the $\hat{D}_2$ polytope.

- It can be immediately verified that there are 10 deformed realisations of the $\hat{D}_3$ polytope. For a generic $n$, the number of such realisations is $n + 1 + (2n)\left[\frac{n}{2}\right]$ for $n \geq 3$.

Let $\Omega_n^{1-\mathrm{L}}$ be the planar scattering form on $\mathbb{R}^{n(n+1)}$ which is defined by the combinatorial $\hat{D}_n$ polytope. We now claim that the pull back of this form on the deformed realisations $\hat{D}_n^{(M,N)}$ of $\hat{D}_n$ generate 1-loop integrand of the S-matrix where all the channels have at most one massive propagator.

**Lemma 6.2.** Let $\Omega_n^{1-\mathrm{L}}$ be the d-ln form on the embedding space which is defined by $\hat{D}_n$ combinatorial polytope. Let $\omega_n^{1-\mathrm{L}} := \Omega_n^{1-\mathrm{L}}|_{\hat{D}_n^{(M,N)}}$ denote it's pull-back on the deformed realisation $\hat{D}_n^{1-\mathrm{L}}$.

Then there exists a set of deformation parameters as parametrized in eqn.(6.29, 6.30, 6.31), such that a weighted sum over $\mathcal{S}_{MN}$ is the n-point (1 loop) integrand at $O(\lambda_2^2)$.

$$
\omega_n^{1-\mathrm{L}} := \sum_{(M,N) \in \mathcal{PC}_n} a_{MN}\,\Omega_n^{1-\mathrm{L}}|_{\hat{D}_n^{(M,N)}}
\tag{6.27}
$$

where the weights are defined as follows.

$$
\begin{aligned}
a_{mk} &= \tfrac{1}{|m-k|} \text{ if } 1 \leq m < k - 1 < n - 1 \\
&= 1 \text{ otherwise}
\end{aligned}
\tag{6.28}
$$

Then $\omega_n^{1-\mathrm{L}}$ is the S-matrix for the interaction $V(\phi_1, \phi_2)$ in eqn.(5.1) upto order $\lambda_2^2$. That is, it contains sum over all the 1 loop integrands in this theory in which at-most one propagator is massive.

---

[11]In fact, just as in the case of tree-level S-matrix, each of these deformed realisations can be thought of as a convex realisation of a combinatorial polytope that we call colorful $\hat{D}_n$ polytope : A colorful $\hat{D}_n^{\mathcal{S}_{IJ}}$ polytope is the $\hat{D}_n$ polytope in which all the co-dimension one facets which correspond to 1-partial pseudo triangulation in which a chord in $\mathcal{S}_{IJ}$ is absent, are red while the remaining facets are black. The combinatorial colorful polytope will not be used in what follows and hence we do not analyse it any further in this paper.

The the deformation parameters for which the (sum over) canonical forms is the 1-loop integrand are related to the couplings as follows.

If the seed is a linear chord $(m, k)$ with $1 \leq m \leq n$ and $m + 2 \leq k \leq m + (n-1)$ then let,

$$\alpha_{ij}^{m,n} = \frac{\lambda_1^2}{\lambda_2^2} \, \forall \, (i, j) \in \mathcal{S}_{mn} \tag{6.29}$$

If the deformation seed is $(1, \bar{1})$ then let,

$$\alpha_{i,\bar{i}}^{1,\bar{1}} = \frac{\lambda_1}{\lambda_3} \tag{6.30}$$

And finallly if the deformation seed is $(i, 0)$ then,

$$\begin{aligned} \alpha &= \frac{\lambda_1^2}{\lambda_2^2} \\ \beta &= \frac{\lambda_2^2}{\lambda_1 \lambda_3} \end{aligned} \tag{6.31}$$

*Proof.* If the seed is a linear chord $(i, j)$ then any propagator involving loop momenta or $X_{i\bar{i}}$ is massless. In this case, a moment of meditation will convince the reader that,

$$\sum_{i=1}^{n} \sum_{j=i+2}^{i+n} \frac{1}{j-i} \, \Omega_n^{\text{1-L}}|_{\hat{D}_n^{(i,j)}} + \Omega_n^{\text{1-L}}|_{\hat{D}_n^{(1,n)}} \tag{6.32}$$

is sum over all the Feynman diagrams in which only at most one propagator (dual to a chord in $\mathcal{S}_{ij}$) is massive. In eqn.(6.32), the range of $j$ is $\{3 \ldots, n\bar{1}, \ldots, \bar{n}\}$. That is $n + i = \bar{i} \, \forall \, 1 \leq i \leq n$ in the sum. The coefficient of each term is determined by following the proof of lemma (4.1) in [17].

If the seed is $(i, 0)$ then the deformed realisation has precisely two co-dimension one facets that correspond to the massive pole. Namely $Y_i = \tilde{Y}_i = m^2$. In this case, the pull back of the canonical form on $\hat{D}_n^{(i,0)}$ can be computed as follows.

Let $PT_1, PT_2$ be two pseudo-triangulations which are dual to two Feynman graphs $PT_1^\star, PT_2^\star$ respectively. Let $(i, 0)$ be a chord in $PT_1$ such that $(i, \bar{i}) \notin PT_1$.

Let $PT_2$ be one of the two types of pseudo-triangulations obtained by a single mutation from $PT_1$.

$$(i, 0), (i, \bar{i}) \in PT_2 \text{ or } (i, 0) \notin PT_2 \tag{6.33}$$

In both of these cases, the residue of the (pullback of) canonical form on the deformed realisation $\hat{D}_n^{(i,0)}$ on vertices $PT_1, PT_2$ will correspond to two terms in the S-matrix integrand if and only if

$$\prod_{e \in PT_1} \alpha_e \prod_{t \in PT_1} \lambda_t = \prod_{e \in PT_2} \alpha_e \prod_{t \in PT_2} \lambda_t \tag{6.34}$$

If **(1)** $PT_2$ is a pseudo-triangulation that is obtained by a single mutation on $PT_1$ which takes $(k, 0)$ for some $k$ to $(i, \bar{i})$ then this constraint reduces to,

$$\alpha \, \lambda_2^2 \; = \; \beta \, \lambda_3 \, \lambda_1 \tag{6.35}$$

If **(2)** $PT_2$ does not contain $(i, 0)$ then the constraint reduces to,

$$\alpha \, \lambda_2^2 \; = \; \lambda_1^2 \tag{6.36}$$

Hence in this case we see that by choosing the deformation parameters as in equations (6.29), (6.30), (6.31), the pull-back $\Omega_n^{1-L}|_{\hat{D}_n^{(i,0)}}$ is a sum over all the Feynman graphs of $\phi^3$ planar 1-loop integrand with at most one massive propagator.

$\square$

# 7    Discussion

The Positive geometry program for S-matrix is built around a "universal" premise of discovering a class of closed convex polytopes in the kinematic space of Mandelstam invariants. These geometries are a specific realisations of a class of combinatorial polytopes which are intimately tied to dissections of an abstract $n$-gon. Perhaps one of the most striking aspects of the program is how the convex realisations of these combinatorial polytopes, e.g. the associahedron which are relevant for the S-matrix program arise due to a completely independent link between the combinatorial objects and the theory of cluster (or more generally gentle) algebras.

Apart from Poincare invariance, no postulate (from the postulates of the analytic S-matrix program of 60's) is assumed in finding the positive geometries whose associated canonical forms define S-matrix of local and unitary QFTs.

Although the ABHY realisation of the associahedron was directly in the kinematic space, latter works such as [8, 9] proved how the same realisations can be thought of as polytopal realisations of $g$-vector fans associated to finitely generated cluster algebras (or more generally gentle algebras) and this realisation is in an Euclidean space (called embedding space in this paper to distinguish it from the kinematic space) with co-ordinates $\kappa_{ij}$ labelled by the dissections of the polygon. If we directly identify the co-ordinates $\kappa_{ij}$ of the embedding space with $X_{ij}$, then, as was shown in [2] and a number of subsequent works [10, 11, 13] that we get positive geometries for S-matrix associated to massless scalar theories.

However, this trivial identification of the co-ordinate system of the embedding space $\{ \kappa_{ij} \}$ and the planar kinematic basis $\{ X_{ij} \}$ is an additional input which we relax in this paper. And this results in several interesting consequences for both, the positive geometry program and the CHY (Cachazo, He and Yuan) formula for scatering amplitudes.

More in detail, we propose to classify the (non-trivial) linear diffeomorphisms between the embedding space and the kinematic space such that the ABHY realisation of an associahedron that is deformed in the kinematic space, still defines S-matrix of *some* unitary

local QFTs. Although the complete classification is far beyond the scope of this paper, we gave several classes of examples of the deformed realisations of the associahedron which are positive geometry of tree-level S matrix for some QFT.

We then showed that in some simple cases, even the converse is true. Namely, for Lagrangian involving more than one fields and more than one coupling, there does exist deformed realisation of the associahedron which is the sought after positive geometry.

Hence the universality in positive geometry program is perhaps even more far reaching then previously thought : The ABHY realisation and the planar scattering form appear to be the fundamental objects which when "viewed" in different co-ordinates in the kinematic space generate S-matrix of different theories with cubic couplings.

If we fix the external particles to be massless (or of the same mass $M$), then we now have the following result for tree-level S-matrix as well as the S-matrix integrand at one loop :

Let $M_n^{(0)/(1)}$ denote the tree level (resp. 1-loop integrand of) S matrix of the Lagrangian $L(\phi_1, \phi_2)$ in eqn.(6.11) which is sum over all the Feynman diagrams that include no massive propagator, one massive propagator or no massless propagator.

And let us denote the $(n-3)$ dimensional ABHY assodiahedron or a $n$ dimensional AHST $\hat{D}$ polytope collectively as $\mathcal{PL}_n$. Let $\omega(\mathcal{PL}_n)$ be an abstract notation for the pull-back of the planar scattering form (in tree-level or 1-loop case) on the convex realisation $\mathcal{PL}_n$.

Then, there exists a set of $S_{\mathcal{G}}^{(0)/(1)}$ of $\mathcal{G}$ linear maps such that

$$\sum_{\mathcal{G} \in S_{\mathcal{G}}^{(0)/(1)}} a_{\mathcal{G}} \, \omega(\mathcal{G} \cdot \mathcal{PL}_n) = M_n^{(0)/(1)} \tag{7.1}$$

The set $S_{\mathcal{G}}^{(0)/(1)}$ is of course different in the tree-level and one-loop case.

Non-trivial maps between the embedding space and $\mathcal{K}_n$ shed an interesting light on the CHY scattering equations, which in the literature so far are solely determined in terms of the external kinematics.

As we proved however, the CHY parke-taylor form can in fact be pushed-forward to generate the S-matrix form on the deformed realisations of the associahedron, where the push-forward map is defined by the deformed scattering equations. The universality of CHY scattering equations for given external kinematics then is really the universality of the diffeomorphism between the CHY moduli space $\overline{\mathcal{M}}_{0,n}(\mathbb{R})$ and the ABHY realisation in the embedding space. It is the CHY scattering equations, which when composed with the diffeomorphism from embedding space to $\mathcal{K}_n$ push-forward the Parke-Taylor form to tree-level amplitudes in an entire class of multi scalar QFTs with cubic coupling.

Although our investigation has been rather preliminary, several interesting questions already emerge out of this work. We list just two of them.

- Can we find positive geometries for eqn.(6.11) to arbitrary orders in the coupling?

- The deformation maps considered in this paper are diagonal subgroups $\mathcal{G} \subset GL(\frac{n(n-3)}{2}, R) \times R^{\frac{n(n-3)}{2}}$. Can one consider more general deformation maps which have off-diagonal en-

tries? And can the positive geometries account for "non-planar" channels via such maps?

With regards to the second question, we offer a rather wild speculation.

Let us consider $n = 5$ scattering in massless $\phi^3$ theory. We will now show that there exists a deformation map such that it's action on the ABHY associahedron maps to a polytope in $\mathcal{K}_5$ all of whose vertices correspond to non-planar channels.[12]

We consider the following map from $\mathcal{K}_5$ (co-ordinatized by $X_{ij}$) to the embedding space (co-ordinatized by $\kappa_{ij}$).

$$\begin{pmatrix} \kappa_{13} \\ \kappa_{14} \\ \kappa_{24} \\ \kappa_{25} \\ \kappa_{35} \end{pmatrix} = \begin{pmatrix} -1 & 1 & -1 & 0 & 0 \\ 0 & -1 & 1 & -1 & 0 \\ 0 & 0 & -1 & 1 & -1 \\ -1 & 0 & 0 & -1 & 1 \\ 1 & -1 & 0 & 0 & -1 \end{pmatrix} \begin{pmatrix} X_{13} \\ X_{14} \\ X_{24} \\ X_{25} \\ X_{35} \end{pmatrix} \tag{7.2}$$

The deformed realisation of the ABHY associahedron in the embedding space can be drawn as a polytope by considering the 2-dimensional plane given by $\tilde{s}_{ij} = c_{ij}$ and a positive wedge. We denote this polytope as $\mathcal{NCP}_2$ in the kinematic space. There exists a choice of $c_{ij}$ for which this realisation is convex and its shape in the $(X_{13}, X_{14})$ plane is shown in fig. (8) below.

Vertices of $\mathcal{NCP}_2$ are co-ordinatized by following points in the $\tilde{s}_{ij} = c_{ij}$ hyper-plane.

$$\{ (s_{13} = 0, s_{24} = 0), (s_{13} = 0, s_{25} = 0), (s_{14} = 0, s_{25} = 0), (s_{14} = 0, s_{35} = 0), (s_{24} = 0, s_{35} = 0) \} \tag{7.3}$$

In this simple example, the pull-back of the planar scattering form on $\mathcal{NCP}_2$ can be written as,

$$\omega_{n=5}|_{\mathcal{NCP}_2} = \left[ \frac{1}{s_{13}s_{24}} + \frac{1}{s_{13}s_{25}} + \frac{1}{s_{14}s_{25}} + \frac{1}{s_{14}s_{35}} + \frac{1}{s_{24}s_{35}} \right] ds_{13} \wedge ds_{24} \tag{7.4}$$

where $s_{13} = X_{13} + X_{24} - X_{14}$, $s_{24} = X_{24} + X_{35} - X_{25}$. Although this pull-back can not be naturally associated to a volume of the dual polytope projected in one of the co-dimension three planes in $\mathcal{K}_5$, we do see that the rational function that enters in the formula is a sum over 5 non-planner channels in massless $\phi^3$ theory.

Expressing $\tilde{s}_{ij} = -c_{ij}$ (with reference $(13, 14)$) and writing them in terms of $X_{ij}$ we can express $(X_{24}, X_{35}, X_{25})$ in terms of $(X_{13}, X_{14})$ and write,

$$ds_{13} = -\frac{3}{8} dX_{14} + \frac{1}{8} dX_{13} \tag{7.5}$$

$$ds_{14} = \frac{1}{4} dX_{13} + \frac{1}{4} dX_{14} \tag{7.6}$$

---

[12]We note that permutahedron has all but one channel non-planar. However a $d$ dimensional permutahedron is not an associahedron for $d > 1$ and hence the permutahedron in the kinematic space can not be thought of as a deformed realisation of the associahedron.

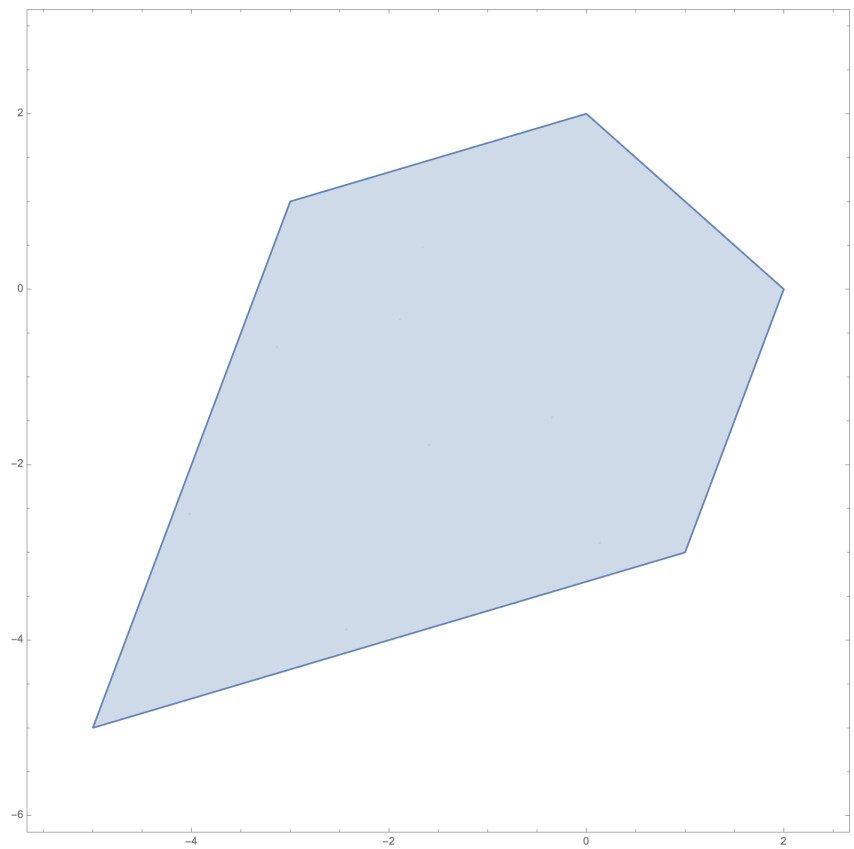

**Figure 8**: $\mathcal{NCP}_2$ in $(X_{13}, X_{14})$ plane

Hence

$$ds_{13} \wedge ds_{14} \;=\; \frac{1}{16} \, dX_{14} \wedge dX_{13} \tag{7.7}$$

Hence we see that the rational function that enters in the formula is proportional to the sum over 5 non-planner channels in massless $\phi^3$ theory. The co-efficient $\frac{1}{16}$ is rather mysterious and is an obstruction to interpreting the pull-back as a (partial) amplitude consisting of the 5 non-planar channels.

Hence at this stage, it is too early to draw conclusive lessons from this single example, or indeed to even obtain the full tree-level scattering 5 point amplitude of massless uncolored scalars. However this example hints at a possibility that non-planar channels maybe "accessible" via space of linear maps from the kinematic space to the embedding space in which ABHY associahedron is located.

We hope to come back to these questions in the future.

## Acknowledgement

We are deeply thankful to Nima Arkani-Hamed for several stimulating discussions, penetrating questions as well as his constant encouragement. We also thank him for explaining the construction of the 1 loop kinematic space as well as $\hat{D}_n$ polytope prior to the publication of [4]. We thank Pinaki Banerjee, Siddharth Prabhu, Prashanth Raman and Pushkal Srivastava for discussions. AL would like to thank TIFR string group for their hospitality where some of these results were presented in the Quantum Space-time Seminar.

## A   Number of possible Couplings $N_c$ in section 4.3

**Lemma A.1.** The number of independent couplings $N_c$ in the n point is given by the following formula.

$$
\begin{aligned}
N_c &= [(\lfloor \tfrac{n}{2} \rfloor - 1) + (\lfloor \tfrac{n}{2} \rfloor - 2)] + [(\lfloor \tfrac{n}{2} \rfloor - 4) + (\lfloor \tfrac{n}{2} \rfloor - 5)] + \ldots \text{ if } \lfloor \tfrac{n}{2} \rfloor \text{ is even} \\
&= [(\lfloor \tfrac{n}{2} \rfloor - 1) + (\lfloor \tfrac{n}{2} \rfloor - 2)] + [(\lfloor \tfrac{n}{2} \rfloor - 4) + (\lfloor \tfrac{n}{2} \rfloor - 5)] + \ldots \text{ if } \lfloor \tfrac{n}{2} \rfloor \text{ is odd}
\end{aligned}
\tag{A.1}
$$

where the dots indicate that the last terms in the sum is either $(4+3)\,1$ or $(6+5),\,(3+2)$. Note that each pair in the sum is such that difference between the second term in $k$-th pair and the first term is $k + 1$th pair is two.

*Proof.* The proof is a simple exercise in combinatorics. Let us first consider the case when $\frac{n}{2} =$ Even. Let us use a "dual" index $I, J, K, \ldots$ to denote a chord between two vertices $i, j$ with $I = |j - i|$. Hence a scalar field with mass $m_{ij}$. $|j - i|$ will be denoted as $\phi_I \,|\, I = j - i$.

Hence we have a spectrum of $\frac{n}{2}$ fields labelled as $\phi_I \,|\, I \in \{1, \ldots, \frac{n}{2}\}$. Let us denote the coupling between $\phi_I$, $\phi_J$, $\phi_K$ as $\lambda_{IJK}$. It is immediate that the list of *independent* couplings can be enumerated in the following disjoint sets.

$$
\begin{aligned}
\text{Set} =& \\
\{\lambda_{112}, \ldots,\ \lambda_{1, \frac{n}{2}-1, \frac{n}{2}-1}\} & \{\lambda_{224}, \ldots,\ \lambda_{2, \frac{n}{2}-2, \frac{n}{2}-1}\} \ldots \\
& (\{\lambda_{\lfloor \frac{n}{3} \rfloor, \lfloor \frac{n}{3} \rfloor, \lfloor \frac{n}{3} \rfloor}\} \text{ or } \{\lambda_{\lfloor \frac{n}{3} \rfloor, \lfloor \frac{n}{3} \rfloor, \lfloor \frac{n}{3} \rfloor + 1}\} \text{ or } \{\lambda_{\lfloor \frac{n}{3} \rfloor, \lfloor \frac{n}{3} \rfloor + 1, \lfloor \frac{n}{3} \rfloor + 1}\})
\end{aligned}
\tag{A.2}
$$

Cardinality of the set defined above equals $N_c$ defined above. This completes the proof.   □

## B   Proof of lemma 4.2

In this appendix we prove the lemma 4.2.

**Lemma B.1.** The canonical form $\Omega_{n-3}|_{A^d_{n-3}}$ associated to $A^d_{n-3}$ is a tree-level color ordered S-matrix for an interacting QFT involving $\lfloor \frac{n}{2} \rfloor + 1$ species of bi-adjoint scalars with masses $m_I \,|\, I = 1, \ldots, \lfloor \frac{n}{2} \rfloor$ with the following constraints on the couplings.

- All the couplings $\lambda_{IJK}$ are non-zero if and only if $I + J = K \bmod n$ and

- The number of relations this set of couplings have to satisfy equals the number $\mathcal{C}_n$ of 3-cycles $c_{IJK}$ in the colored dissection quiver where $I, J, K > 2$ modulo $n$.

*Proof.*

Each coupling $\lambda_{IJK}$ corresponds to a triangle with the three edges of the triangle colored by indices $I, J, K$ respectively. Hence $I + J = K$ modulo $n$. No such couplings can be zero as the residue of the canonical form is non vanishing on all vertices of the associahedron. This proves the first statement.

Before proving the second statement, let us recall that the number of independent $\lambda_{IJK}$ is same as the number of distinct colored triangles which can be drawn using $\lfloor \frac{n}{2} \rfloor$ colored edges corresponding to spectrum of massive fields and the un-coloured external edge that corresponds to the massless field.

Let us consider two terms in $\Omega_{n-3}|_{A^d_{n-3}}$ which correspond to the two triangulations, $T_0 = \{\, 13,\, 14,\, \ldots,\, 1n{-}1 \,\}$ and $T_i = \{\, 13,\, 14,\, \ldots,\, (1, i{-}1),\, (i{-}1, i{+}1),\, (1, i{+}1),\, \ldots,\, 1n{-}1 \,\}$. The ratio of the residues of the form evaluated on the corresponding vertices of $A^d_{n-3}$ equals $\frac{\alpha_{13}}{\alpha_{1i}}$. This implies that ratio of the co-efficient multiplying $\frac{1}{\prod_{i=3}^{n-1} \tilde{X}_{1i}}$, with co-efficient of

$$\frac{1}{\prod_{m_1=3}^{j_1-1} X_{1m_1} \left( X_{j_1-1,j_1+1}\, X_{1,j_1+1} \right) \prod_{m_2=j_1+1}^{j_2-1} X_{1m_2} \left( X_{j_2-1,j_2+1}\, X_{1,j_2+1} \right) \ldots X_{1,n-1}}$$

equals $\left( \frac{\alpha_{13}}{\alpha_{1,j_1}} \frac{\alpha_{13}}{\alpha_{1,j_2}} \ldots \right)$. That is, up to an overall scaling a channel in which all the closed loops $\{\, (ij), (jk), (ki) \,\}$ have at least one edge of length 2 modulo $n$ are generated from the ratios $\{\, \frac{\alpha_{13}}{\alpha_{1i}} \,\}$.

Now consider a vertex of the associahedron corresponding to a triangulation $T$, that contains precisely one closed loop $I, J, K > 2$. The residue of $\Omega_{n-3}$ evaluated on this vertex must satisfy the constraint,

$$\prod_{(ij) \in T} \alpha_{ij} \prod_{\triangle \in T^\star} \lambda_\triangle = \prod_{(ij) \in T_0} \alpha_{ij} \prod_{\triangle \in T_0^\star} \lambda_\triangle \tag{B.1}$$

where $\lambda_\triangle$ contains precisely one coupling $\lambda_0$ with $I, J, K > 2$. As the deformation parameters have been mapped onto the couplings $\lambda_{IJK}$ where atleast one of the indices equals 2, eqn.(B.1) is a constraint on $\lambda_0$. This proves that $\Omega_{n-3}|_{A^d_{n-3}}$ is the $n$ point amplitude in a $\lfloor \frac{n}{2} \rfloor$-biadjoint cubic theory where the set of all couplings satisfy $\mathcal{C}_n$ relations. $\square$

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
