# Peer review of "Towards Positive Geometries of Massive Scalar field theories"

_SciPost Physics_

## Round 1 · Referee Report · Anonymous · 2022-12-9

Report

The article is interesting with novel ideas and sufficent advancement of current ideas. Except for spelling mistakes and grammatical errors, the manuscript seems suitable for publishing.

---

## Round 1 · Referee Report · Anonymous · 2023-1-18

Strengths

1- This paper advances our understanding of the positive geometry description of scalar interactions, a crucial issue within this field of study.

2- The authors give many non-trivial examples of matching between the canonical form of the deformed ABHY associahedra and some scattering amplitudes in perturbation theory.

Weaknesses

1 -This work does not examine the extent to which the new geometric description, or the deformed scattering equations, enhance our comprehension of the characteristics of the scattering amplitude or improve our capability to calculate them.

2-The extent of the validity of the results obtained is not always clear.

3- The references to the literature could be improved.

Report

The paper addresses the study of deformations of the associahedron geometry in the context of scattering amplitudes in planar phi^3 theory. The authors introduce new deformed associahedron spaces that are related to the original associahedron through an affine transformation that acts on the dual kinematic space. They propose that the same deformation applied to the scattering equation should result in a map between the world-sheet associahedron and the deformed ABHY associahedron. The authors also find a class of amplitudes involving multi-field cubic interactions that can be obtained as the canonical form of deformed associahedra. Some of these results are then generalized at 1-loop level, introducing new ideas.

The results of this paper are innovative and original, and have the potential to meet the acceptance standards of this journal with some revisions on technical points as advised by the referee.

Requested changes

There are a few points that I would like to be clarified:

1- In Section 2, it is stated that the generic propagator of a planar amplitude is in the form of $\frac{1}{X_{ij}-m_{ij}}$. However, the general form should be $\frac{1}{X_{ij}-m^2_k}$, as the mass of the propagating particle is independent of the labels $i$ and $j$ and a square on $m_{ij}$ is missing.

2- It is my understanding that the deformed associahedron can be defined as the image of the ABHY associahedron under the affine map $\phi \ :\ k_{ij}=\alpha_{ij}(X_{ij}-m^2_{ij})$. As discussed in reference [1703.04541], this implies that the pushforward of the canonical form of the associahedron through $\phi$ is the canonical form of the deformed associahedra. Since $\phi$ is a bijection the pushforward is trivial and the amplitude described by the deformed associahedron is, therefore, given by the massless $\phi$ amplitude with the propagators $X_{ij}$ replaced by $\alpha_{ij}(X_{ij}-m^2_{ij})$. There is no need to invoke the CHY deformed scattering equation. I think this should be commented on.

3- After equation (4.7), it is claimed that the canonical form of the deformed amplituhedron defined in (4.1) gives the tree-level amplitudes for a Lagrangian with interaction $\phi_2^3+\phi_1^2\phi_2+\phi_1\phi_2^2+m\phi_2^2$. However, it is not clearly stated here that you restrict to massless external kinematics. Additionally, starting at 6-particle amplitudes with massless kinematics contain both massive and massless propagators, while the canonical form of the deformed amplituhedron only has massive poles. Are you restricting to just massive propagators? If so, what is the physical reason to consider just this contribution?

4- Also in section 4.2 it looks like you are computing just a contribution to the amplitude, which corresponds to the sum over Feynman diagrams with only massive propagators. In the abstract, it it's written that only the $\lambda_2^2$ contribution is considered, but this is not reported in section 4. These points should be made clearer.

5-In section 5, the S-matrix of a theory with 2 massless scalars is considered. Is my impression that you consider $\phi_2$ to be massive and allows only for massless external kinematics, but this is not stated.

6- The motivations behind the definition in eq (5.2) are not clear.

7- What does the superindex $co$ mean in equations (5.9) and (5.11) ? Was it $M_n^{(2)}$ instead?

As a suggestion, the authors may want to consider:

1- Adding explicit amplitude formulas, such as eq (5.8), for the 6-particle amplitudes in sections 4.1 and 4.2, to help the readers familiarize themself with multi-field amplitudes.

2- Including more references on recent developments in the field of positive geometries, such as the computation of pushforwards via scattering equations of canonical forms, weighted positive geometries, and the work of Dolon and Goddard on off-shell CHY, which are referred to but not explicitly cited your manuscript.

3- Citing in which paper equation (2.2) has been derived, as it is more general than the one proposed in the original ABHY construction of the associahedron.

4- Proofreading the text, as there are many small typos and grammatical errors present. This can be easily done with the help of appropriate software.

---

## Editorial Decision

awaiting_resubmission